# Long-term balancing selection drives evolution of immunity genes in *Capsella*

**Daniel Koenig[1][†]\*, Jörg Hagmann[1][‡], Rachel Li[1][§], Felix Bemm[1][#], Tanja Slotte[2], Barbara Neuffer[3], Stephen I Wright[4], Detlef Weigel[1]\***

[1]Department of Molecular Biology, Max Planck Institute for Developmental Biology, Tübingen, Germany; [2]Department of Ecology,Environment, and Plant Sciences, Stockholm University, Stockholm, Sweden; [3]Department of Biology, University of Osnabrück, Osnabrück, Germany; [4]Department of Ecology and Evolutionary Biology, University of Toronto, Toronto, Canada

**Abstract** Genetic drift is expected to remove polymorphism from populations over long periods of time, with the rate of polymorphism loss being accelerated when species experience strong reductions in population size. Adaptive forces that maintain genetic variation in populations, or balancing selection, might counteract this process. To understand the extent to which natural selection can drive the retention of genetic diversity, we document genomic variability after two parallel species-wide bottlenecks in the genus *Capsella*. We find that ancestral variation preferentially persists at immunity related loci, and that the same collection of alleles has been maintained in different lineages that have been separated for several million years. By reconstructing the evolution of the disease-related locus *MLO2b*, we find that divergence between ancient haplotypes can be obscured by referenced based re-sequencing methods, and that trans-specific alleles can encode substantially diverged protein sequences. Our data point to long-term balancing selection as an important factor shaping the genetics of immune systems in plants and as the predominant driver of genomic variability after a population bottleneck.
DOI: https://doi.org/10.7554/eLife.43606.001

**\*For correspondence:**
dkoenig@ucr.edu (DK);
weigel.elife@gmail.com (DW)

**Present address:** [†]Department of Botany and Plant Sciences, University of California, Riverside, United States; [‡]Computomics GmbH, Tübingen, Germany; [§]Berkeley Brewing Science, Oakland, United States; [#]KWS SE, Einbeck, Germany

## Introduction

Balancing selection describes the suite of adaptive forces that maintain genetic variation for longer than expected by random chance. It can have many causes, including heterozygous advantage, negative frequency-dependent selection, and environmental heterogeneity in space and time. The unifying characteristic of these situations is that the turnover of alleles is slowed, resulting in increased diversity at linked sites (*Charlesworth, 2006*). In principle, it should be simple to detect the resulting footprints of increased coalescence times surrounding balanced sites (*Tellier et al., 2014*), and many candidates have been identified using diverse methodology (*Fijarczyk and Babik, 2015*). However, balanced alleles will be stochastically lost over long time spans, suggesting that most balanced polymorphism is short lived (*Fijarczyk and Babik, 2015*).

The strongest evidence for balancing selection comes from systems in which alleles are maintained in lineages that are reproductively isolated and that have separated millions of years ago, resulting in trans-specific alleles with diagnostic trans-specific single nucleotide polymorphisms (tsSNPs). A few, well known genes fit this paradigm: the self-incompatibility loci of plants (*Vekemans and Slatkin, 1994*), mating-type loci of fungi (*Wu et al., 1998*), and the major histocompatibility complex (MHC) and ABO blood group loci in vertebrates (*McConnell et al., 1988*; *Mayer et al., 1988*; *Lawlor et al., 1988*; *Watkins et al., 1990*; *Ségurel et al., 2012*). Additional candidates have been proposed by comparing genome sequences from populations of humans and chimpanzees, and from populations of multiple *Arabidopsis* species. These efforts have revealed six

**eLife digest** *Capsella rubella* is a small plant that is found in southern and western Europe. This plant is young in evolutionary terms: it is thought to have emerged less than 200,000 years ago from a small group of plants belonging to an older species known as *Capsella grandiflora*.

Individuals of the same species may carry alternative versions of the same genes – known as alleles – and the total number of alleles present in a population is referred to as genetic diversity. When a few individuals form a new species, the gene pool and the genetic diversity in the new species is initially much lower than in the ancestral species, which may make the new species less robust to fluctuations in the environment. For example, alternative versions of a gene might be preferable in hot or cold climates, and loss of one of these versions would limit the species' ability to survive in both climates.

A mechanism known as balancing selection can maintain various alleles in a species, even if the population is very small. However, it was not clear how common long-lasting balancing selection was after a species had split. To address this question, Koenig et al. assembled collections of wild *C. rubella* and *C. grandiflora* plants and sequenced their genomes in search of alleles that were shared between individuals of the two species.

The analysis found not just a few, but thousands of examples where the same genetic differences had been maintained in both *C. rubella* and *C. grandiflora*. Some of these allele pairs were also shared with individuals of a third species of *Capsella* that had split from *C. rubella* and *C. grandiflora* over a million years ago. The shared alleles did not occur randomly in the genome; genes involved in immune responses were far more likely to be targets of balancing selection than other types of genes.

These findings indicate that there is strong balancing selection to maintain different alleles of immunity genes in wild populations of plants, and that some of this diversity can be maintained over hundreds of thousands, if not millions of years. The strategy developed by Koenig et al. may help to identify new versions of immunity genes from wild relatives of crop plants that could be used to combat crop diseases.

DOI: https://doi.org/10.7554/eLife.43606.002

loci in primates (*Leffler et al., 2013b*; *Teixeira et al., 2015*) and up to 129 loci, that were identified by at least two shared SNPs each, in *Arabidopsis* (*Novikova et al., 2016*; *Bechsgaard et al., 2017*), as potential targets of long-term balancing selection and/or introgression. In both systems, genes involved in host–pathogen interactions were enriched, which in *Arabidopsis* is consistent with previous findings that several disease resistance loci appear to be under balancing selection in this species, based on the analysis of individual genes (*Huard-Chauveau et al., 2013*; *Botella, 1998*; *Caicedo et al., 1999*; *Noel, 1999*; *Stahl et al., 1999*; *Tian et al., 2002*; *Bakker, 2006*; *Rose et al., 2004*; *Todesco et al., 2010*). However, even with the ability to conduct whole-genome scans for balancing selection in *A. thaliana*, the total number of examples with robust evidence across species remains small (*Cao et al., 2011*; *1001 Genomes Consortium, 2016*).

One explanation for this paucity of evidence for pervasive and stable balancing selection is that cases of long-term maintenance of alleles are rare. However, there are good reasons to believe that many studies lacked the power to detect the expected effects (*Fijarczyk and Babik, 2015*; *DeGiorgio et al., 2014*). If one requires that alleles have been maintained in species separated by millions of years, then only targets of outstandingly strong selective pressures that remain the same over many millennia can be identified. Furthermore, recombination between deeply coalescing alleles will typically reduce the size of the genomic footprint to very short sequence stretches, thus limiting the opportunity for distinguishing old alleles from recurrent mutations.

We hypothesised that self-fertilizing species provide increased sensitivity to detect balancing selection based on two observations (*Wiuf et al., 2004*; *Wright et al., 2008*). First, self fertilisation greatly reduces the effective rate of recombination, thus potentially expanding the footprint of balancing selection. In addition, the transition to self fertilisation is generally associated with dramatic genome-wide reductions in polymorphism, potentially making it easier to detect outlier loci that retain variation from the outcrossing, more polymorphic ancestor. In this study we sought to assess

how strongly selection acts to maintain genetic diversity in the context of repeated transitions to self fertilisation in the flowering plant genus *Capsella*. Like many plant lineages, the ancestral state of *Capsella* is outcrossing (found in the extant diploid species *C. grandiflora*), but selfing has evolved independently in two diploid species, *C. rubella* and *C. orientalis* (*Figure 1A*)(*Foxe et al., 2009*; *Guo et al., 2009*; *Bachmann et al., 2018*). The genomes of both species exhibit the drastic loss of genetic diversity typical for many selfers (*Figure 1B–C*) (*Guo et al., 2009*; *Foxe et al., 2009*; *St Onge et al., 2011*; *Slotte et al., 2013*; *Brandvain et al., 2013*; *Slotte et al., 2012*). In the younger species, *C. rubella*, loss of genetic diversity was initially thought to have occurred uniformly throughout the entire genome (*Foxe et al., 2009*; *Guo et al., 2009*), but subsequent reports already hinted at some loci having increased diversity (*Gos et al., 2012*; *Brandvain et al., 2013*), motivating the present study.

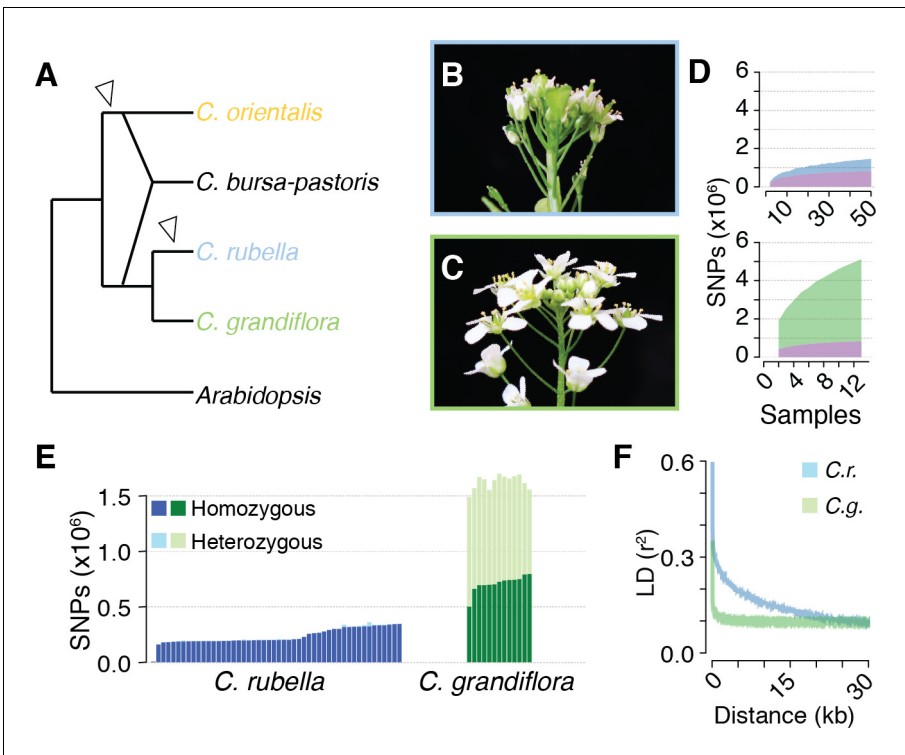

**Figure 1.** Polymorphism discovery in *Capsella*. (A) Diagram of the relationships between *Capsella* species. Arrowheads indicate transitions from outcrossing to self-fertilisation. (B) Inflorescence of *C. rubella* with small flowers. (C) Inflorescence of *C. grandiflora* with large, showy flowers, to attract pollinators. (D) SNP discovery in *C. rubella* (top) and *C. grandiflora* (bottom). Samples were randomly downsampled ten times. Means of segregating transpecific (tsSNPs, purple), species specific in *C. rubella* (ss$_{Cr}$SNPs, blue), and species specific in *C. grandiflora* (ss$_{Cg}$SNPs, green) SNPs. (E) Number of heterozygous (light colours) and homozygous SNP calls (dark colours). (F) Average decay of linkage disequilibrium in *C. grandiflora* (green) and *C. rubella* (blue).
DOI: https://doi.org/10.7554/eLife.43606.003

The following source data and figure supplement are available for figure 1:

**Source data 1.** Sample information.
DOI: https://doi.org/10.7554/eLife.43606.005
**Source data 2.** Diversity and divergence estimates for C.grandiflora and C. rubella.
DOI: https://doi.org/10.7554/eLife.43606.006
**Figure supplement 1.** Map of collections.
DOI: https://doi.org/10.7554/eLife.43606.004

## Results

### Polymorphism discovery in *C. grandiflora* and *C. rubella*

The species *Capsella rubella* is young, only 30,000 to 200,000 years old, and was apparently founded when a small number of *C. grandiflora* individuals became self-compatible (*Foxe et al., 2009*; *Guo et al., 2009*). Previous studies had hinted at unequal retention of *C. grandiflora* alleles across the *C. rubella* genome (*Gos et al., 2012*; *Brandvain et al., 2013*), leading us to analyse this phenomenon systematically by comparing the genomes of 50 *C. rubella* and 13 *C. grandiflora* accessions from throughout each species' range (*Figure 1—figure supplement 1* and *Figure 1—source data 1*). Because the calling of trans-specific SNPs (tsSNPs) is particularly sensitive to mismapping errors in repetitive sequences, we applied a set of stringent filters, resulting in 74% of the *C. rubella* reference genome remaining accessible to base calling in both species, with almost half (47%) of the masked sites in the repeat rich pericentromeric regions. After filtering, there were 5,784,607 SNPs and 883,837 indels. Unless otherwise stated, all subsequent analyses were performed using SNPs. Of these, only 27,852 were fixed between the two species, whereas 824,540 were found in both species (ts$_{CgCr}$SNPs), consistent with the expected sharing of variation between the two species. In addition, 4,291,959 SNPs segregated only in *C. grandiflora* (species-specific SNPs; ss$_{Cg}$SNPs), and 640,256 only in *C. rubella* (ss$_{Cr}$SNPs). Sample rarefaction by subsampling our sequenced accessions indicated that common ss$_{Cr}$SNP and ts$_{CgCr}$SNP discovery was near saturation in our experiment, though additional sampling will continue to uncover rare alleles (*Figure 1D*).

The consequences of selfing are easily seen as a dramatic reduction in genetic diversity in *C. rubella* (*Figure 1—source data 2*), consistent with the previously suggested genetic bottleneck (*Foxe et al., 2009*; *Guo et al., 2009*). As expected from a predominantly selfing species, SNPs segregating in *C. rubella* were much less likely to be heterozygous than those segregating in *C. grandiflora*, though evidence for occasional outcrossing in *C. rubella* is observed in the form of a variable number of heterozygous calls (*Figure 1E*). Selfing is also expected to reduce the effective rate of recombination between segregating polymorphisms. Linkage disequilibrium (LD) decayed, on average, to 0.1 within 5 kb in *C. grandiflora*, while it only reached this value at distances greater than 20 kb in *C. rubella* (*Figure 1F*). Though *C. rubella* is a relatively young species, it exhibits characteristics typical of a predominantly (but not exclusively) self-fertilising species: reduced genetic diversity, reduced observed heterozygosity, and reduced effective recombination rate. This last effect could potentially increase the visibility of signals for balancing selection from linked sites (*Wiuf et al., 2004*).

### *Capsella rubella* demography

The degree of trans-specific allele sharing depends upon the level of gene flow between species, the age of the speciation event, and the demographic history of each resultant species. We first sought to understand how these neutral processes have affected extant polymorphism in *C. grandiflora* and *C. rubella*. We searched for evidence of population structure in our dataset by fitting individual ancestries to different numbers of genetic clusters with ADMIXTURE (*Alexander et al., 2009*) (*Figure 2A* and *Figure 2—figure supplement 1A-B*; *k*-values from 1 to 6). The best fit as determined by the minimum cross-validation error was three clusters, with one including all *C. grandiflora* individuals, and *C. rubella* samples split into two clusters. Principal component (PC) analysis (*Price et al., 2006*) of genetic variation revealed a similar picture, with PC1 separating the two species and PC2 separating the *C. rubella* samples (*Figure 2A*).

*C. rubella* population structure was strongly associated with geography. Samples from western Europe and southeastern Greece were unambiguously assigned to separate groups, while samples from northern and western Greece, near the presumed site of speciation in the current range of *C. grandiflora* (*Hurka and Neuffer, 1997*), showed mixed ancestry (or intermediate assignment to these groups, *Figure 2A–B*). A single *C. rubella* sample from western Europe showed some mixed ancestry. This sample was collected near Gargano National Park on the eastern coast of Italy. The source of its mixed ancestry is unclear, but its proximity to Greece suggests that it may result from ongoing migration across the Adriatic Sea. The general pattern of population structure is consistent with the centre of diversity for *C. rubella* being in northern Greece and a more recent rapid expansion into Western Europe, and agrees with predictions made based on previous, smaller datasets

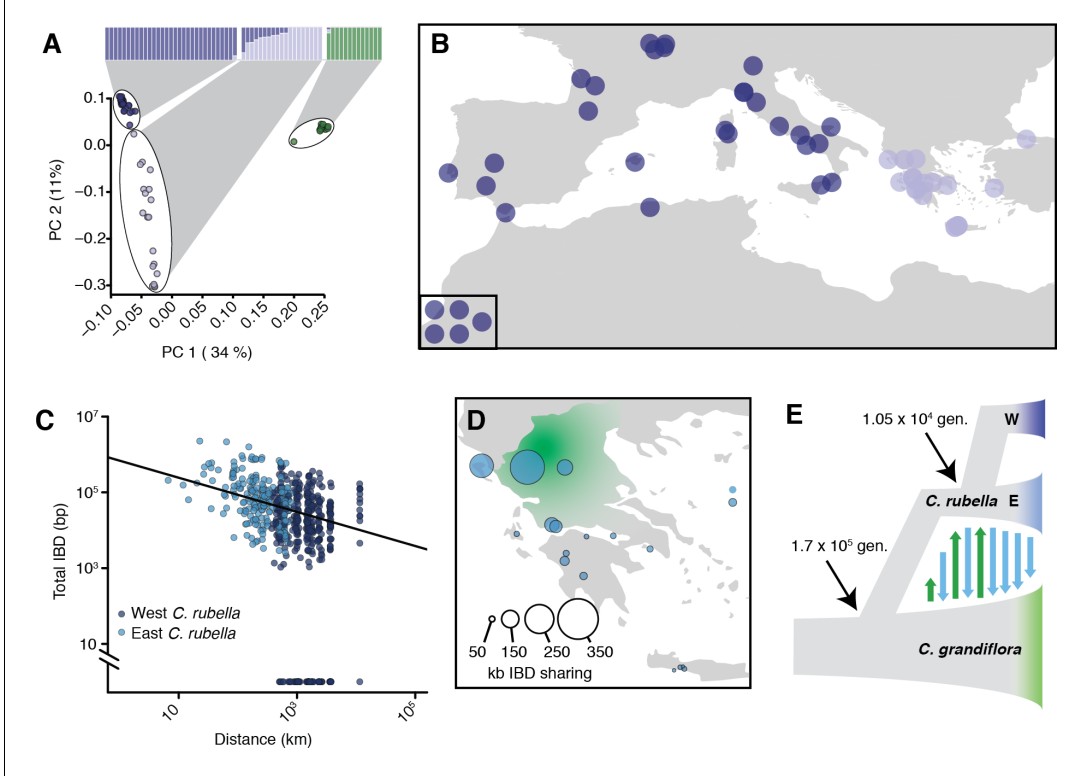

**Figure 2.** Demographic analysis of *C.rubella*. (**A**) Admixture bar graphs (top) and PCA of population structure in *C. grandiflora* (green) and *C. rubella* (blue). The *C. rubella* colours correspond to the sampling locations in (**B**). Inset shows lines from outside Eurasia (Canary Islands and Argentina). (**C**) Pairwise interspecific identity-by-descent (IBD) between *C. grandiflora* and *C. rubella* samples. Comparisons between West *C. rubella* and *C. grandiflora* are in dark blue and E *C. rubella* and *C. grandiflora* in light blue. The minimum segment length threshold was 1 kb, and comparisons without IBD segments (all from the West *C. rubella* population) are at the bottom of the plot. (**D**) Total lengths of interspecific IBD sharing by sample site within the E *C. rubella* population. An approximate distribution of *C. grandiflora* is shown for comparison in green. (**E**) The most likely demographic model of *C. rubella* and *C. grandiflora* evolution as inferred from joint allele frequency spectra by fastsimcoal2. Arrows indicate gene flow.

DOI: https://doi.org/10.7554/eLife.43606.007

The following source data and figure supplements are available for figure 2:

**Source data 1.** D statistics comparing East and West C.rubella populations.
DOI: https://doi.org/10.7554/eLife.43606.010

**Source data 2.** Inferred demographic parameters from fastsimcoal2.
DOI: https://doi.org/10.7554/eLife.43606.011

**Figure supplement 1.** Additional population structure and migration analyses.
DOI: https://doi.org/10.7554/eLife.43606.008

**Figure supplement 2.** Comparison of simulated and observed allele frequency spectra under the best fitting demographic model.
DOI: https://doi.org/10.7554/eLife.43606.009

(*Brandvain et al., 2013*). The observed structure is principally organised by a major geographic barrier, the Adriatic Sea. We therefore separated our samples into into two distinct groups to the west (W) and east (E) of the Adriatic Sea for subsequent analyses.

Because their current ranges overlap, ongoing gene flow between sympatric *C. rubella* and *C. grandiflora* could be a potentially important source of allele sharing between the two species. While a previous study had not found any evidence for such a scenario (*Brandvain et al., 2013*), one of our *C. grandiflora* samples was assigned partial ancestry to the otherwise *C. rubella*-specific clusters, and resided at an intermediate position along PC1 (*Figure 2A*). Furthermore, eastern *C. rubella* individuals, many of which grew in sympatry with *C. grandiflora*, were less differentiated from *C. grandiflora* compared to western *C. rubella* samples along PC1 (*Figure 2A* and *Figure 2—figure supplement 1C-D*). Gene flow between eastern *C. rubella* and *C. grandiflora* was supported by significant genome-wide *D*-statistics for *C. rubella* samples from the *C. grandiflora* range (ABBA-BABA

test; comparing each E individual with the W population) (*Green et al., 2010*; *Durand et al., 2011*), with *D* decreasing as a function of distance from the centre of *C. grandiflora*'s range (*Figure 2—figure supplement 1* and *Figure 2—source data 1*). Because *D* statistics can be sensitive to ancient population structure (*Durand et al., 2011*), we further relied on identity-by-descent (IBD) segments as detected by BEAGLE (*Browning and Browning, 2013*) to identify genomic regions of more recent co-ancestry across these species. The proportion of the genome shared in IBD segments between *C. rubella* and *C. grandiflora* also decreased as a function of distance between samples, and the strongest evidence for recent ancestry was found between *C. grandiflora* individuals and sympatric northern Greek *C. rubella* lines (*Figure 2C–D*). These results indicate that gene flow is ongoing between the species, consistent with interspecific crosses often producing fertile offspring, specifically with *C. rubella* as the paternal parent (*Sicard et al., 2011*; *Rebernig et al., 2015*).

To estimate the magnitude and direction of gene flow and other demographic events that have shaped genetic variation in the two species we used fastsimcoal2 (*Excoffier et al., 2013*) to compare the likelihood of a large number of demographic models given the observed joint site frequency spectrum (*Figure 2E*, *Figure 2—figure supplement 2* and *Figure 2—source data 2*). The best fitting model estimated the split between *C. rubella* and *C. grandiflora* to have occurred 170,000 generations ago, associated with a strong reduction in *C. rubella* population size (to only 2–14 effective chromosomes, or 1–7 individuals). Bidirectional gene flow at a relatively low rate apparently occurred until just over 10,000 generations ago, when *C. rubella* split into the W and E populations, after which gene flow continued only from E *C. rubella* to *C. grandiflora* (*Figure 2E*).

The close timing of the end of gene flow into *C. rubella* and the split into two populations suggests that westward expansion of the *C. rubella* range reduced the opportunity for gene flow from *C. grandiflora*, with potential genetic reinforcement by the development of hybrid incompatibilities (*Sicard et al., 2015*). If we assume an average of 1.3 years per generation as found in the close relative, *A. thaliana* (*Falahati-Anbaran et al., 2014*), which has similar life history and ecology, the population split and the end of introgression from *C. grandiflora* occurred around 13,500 years ago. This date is similar to the spread of agriculture and the end of the last glaciation in Europe (*Walker et al., 2009*), suggesting that *C. rubella*'s success might have been facilitated by one or both of these events.

## Non-random polymorphism sharing after a genetic bottleneck

Our analyses provide dates for the bottleneck and rapid colonisation events that have led to dramatically reduced genetic variation in *C. rubella*. Yet, over half of the segregating variants in *C. rubella* were also found in *C. grandiflora* (*Figure 1D*). Such $ts_{CgCr}$SNPs could originate from independent mutation in each species (identity by state, IBS). Alternatively, they could be the result of introgression after speciation or they could reflect retention of the same alleles since the species split (identity by descent, IBD). Older retained alleles are expected to be found at elevated frequencies relative to the genome-wide average, while younger, recurrent mutations are expected to be rare. We therefore identified ancestral and derived alleles by comparison with the related genus *Arabidopsis*, and then compared the derived allele frequency spectra of $ts_{CgCr}$SNPs and ssSNPs in *Capsella* as a proxy for allele age. We found that $ts_{CgCr}$SNPs are strongly enriched among high-frequency alleles in both *Capsella* species (*Figure 3A*, p-value << 0.0001 in *C. grandiflora* and *C. rubella*, Mann-Whitney U-test). At allele frequencies greater than 0.25 in *C. rubella*, $ts_{CgCr}$SNPs accounted for more than 80% of all variation. These results indicate that $ts_{CgCr}$SNPs are predominantly older alleles that were already present in the common ancestral population of *C. rubella* and *C. grandiflora* or that were introgressed from *C. grandiflora* to *C. rubella* prior to its expansion into western Europe.

The distribution of $ts_{CgCr}$SNPs was uneven across the genome. When compared to $ss_{Cr}$SNPs drawn from the same allele frequency distribution, $ts_{CgCr}$SNPs were less likely to result in nonsynonymous changes (*Figure 3B*, p-value < 0.001, from 1000 jackknife resamples from the same allele frequency distribution), but they were more likely to be in genes (*Figure 3C*). As expected for transpecific haplotype sharing, eighty-three percent of all $ts_{CgCr}$SNPs were in complete LD with at least one other $ts_{CgCr}$SNP in *C. rubella*, and the density of tsSNPs along the genome was highly variable (*Figure 3D–G*). $ts_{CgCr}$SNP density was positively correlated with local genetic diversity in *C. rubella* (and less strongly so with genetic diversity in *C. grandiflora*; *Figure 3F–I* and *Figure 3—figure supplements 2–5*), and negatively correlated with differentiation between the species as measured by $F_{st}$ (*Figure 3J* and *Figure 3—figure supplements 2–5*). The uneven pattern of diversity

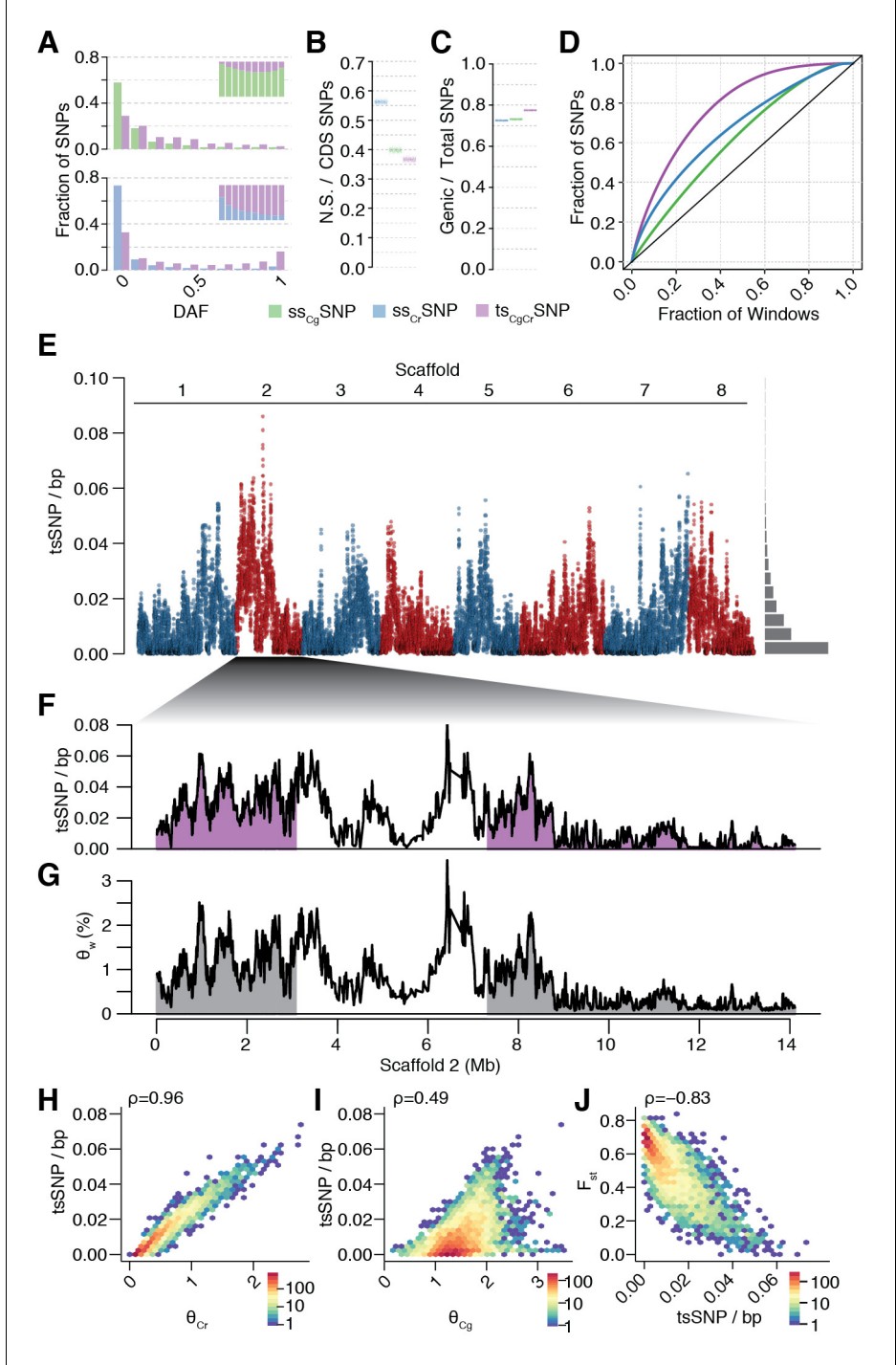

**Figure 3.** Unequal presence of ancestral variation in modern *C.rubella*. (**A**) Derived allele frequency spectra (DAF) of ss$_{Cg}$SNPs (green), ss$_{Cr}$SNPs (blue), and ts$_{CgCr}$SNPs (purple) in *C. grandiflora* (top) and *C. rubella* (bottom). The inset depicts the fraction of alleles that are species or transpecific as a function of derived allele frequency (DAF). (**B**) Fraction of coding (CDS) SNPs that result in non-synonymous changes as a function of SNP sharing. (**C**) Fraction of genic SNPs as a function of SNP sharing. Because SNPs in different classes (ssSNPs, tsSNPs) differ in allele frequency distributions, we normalised by downsampling to comparable frequency spectra. Each bar consists of 1000 points depicting downsampling values. (**D**) 20 kb genomic windows required to cover different fractions of ssSNPs and tsSNPs. The black line corresponds to a completely even distribution of SNPs in the genome. tsSNPs deviate the most from this null distribution. (**E**) ts$_{CgCr}$SNP density in 20 kb windows (5 kb steps) along the eight *Capsella* chromosomes. Histogram on the right shows distribution of values across the entire

*Figure 3 continued on next page*

*Figure 3 continued*

genome. (F) ts$_{CgCr}$SNP density and (G) Watterson's estimator ($\Theta_w$) of genetic diversity along scaffold 2. The repeat dense pericentromeric regions are not filled. (H–J) Correlation of ts$_{CgCr}$SNP density in 20 kb non-overlapping windows with genetic diversity in *C. rubella* (H), genetic diversity in *C. grandiflora* (I), and interspecific F$_{st}$ (J). Spearman's rho is always given on the top left. Only windows with at least 5000 accessible sites in both species were considered.

DOI: https://doi.org/10.7554/eLife.43606.012

The following figure supplements are available for figure 3:

**Figure supplement 1.** Sharing of ts$_{CgCr}$SNPs and ss$_{Cr}$SNPs alleles after colonisation.
DOI: https://doi.org/10.7554/eLife.43606.013

**Figure supplement 2.** Diversity in *C. rubella* and *C. grandiflora* along scaffolds (chromosomes) 1 and 2.
DOI: https://doi.org/10.7554/eLife.43606.014

**Figure supplement 3.** Diversity in *C. rubella* and *C. grandiflora* along scaffolds (chromosomes) 3 and 4.
DOI: https://doi.org/10.7554/eLife.43606.015

**Figure supplement 4.** Diversity in *C. rubella* and *C. grandiflora* along scaffolds (chromosomes) 5 and 6.
DOI: https://doi.org/10.7554/eLife.43606.016

**Figure supplement 5.** Diversity in *C. rubella* and *C. grandiflora* along scaffolds (chromosomes) 7 and 8.
DOI: https://doi.org/10.7554/eLife.43606.017

**Figure supplement 6.** Distribution of diversity in East and West *C. rubella* populations along scaffolds (chromosomes) 1 and 2.
DOI: https://doi.org/10.7554/eLife.43606.018

**Figure supplement 7.** Distribution of diversity in East and West *C. rubella* populations along scaffolds (chromosomes) 3 and 4.
DOI: https://doi.org/10.7554/eLife.43606.019

**Figure supplement 8** Distribution of diversity in East and West *C. rubella* populations along scaffolds (chromosomes) 5 and 6.
DOI: https://doi.org/10.7554/eLife.43606.020

**Figure supplement 9.** Distribution of diversity in East and West *C. rubella* populations along scaffolds (chromosomes) 7 and 8.
DOI: https://doi.org/10.7554/eLife.43606.021

---

was similar in each *C. rubella* subpopulation (*Figure 3—figure supplements 6–9*), indicating that most of the retained polymorphism already segregated prior to colonisation. Thus, most common genetic variation in *C. rubella* is also retained in its outcrossing ancestor, and the rate of retention varies dramatically between genomic regions.

## High density of tsSNPs around immunity-related loci

The observed heterogeneity in shared diversity across the *C. rubella* genome could be a simple consequence of a bottleneck during the transition to selfing. In the simplest scenario, *C. rubella* was founded by a small number of closely related individuals, and stochastic processes during subsequent inbreeding caused random losses of population heterozygosity. A study of genetic variation in bottlenecked populations of the Catalina fox found this exact pattern (*Robinson et al., 2016*). Alternatively, there may be selective maintenance of diversity in specific regions of the genome due to balanced polymorphisms, with contrasting activities of the different alleles. To explore this latter possibility, we tested whether the likelihood of allele sharing was dependent on annotated function of the affected genes. We found that ts$_{CgCr}$SNPs were strongly biased towards genes involved in plant biotic interactions, including defense and immune responses, and also toward pollen-pistil interactions, though less strongly (*Supplementary file 1*, *Figure 4A*). Amongst the top ten enriched Gene Ontology (GO) categories for biological processes were apoptotic process, defense response, innate immune response, programmed cell death, and defensive secondary metabolite production (specifically associated with terpenoids). Of genes annotated with apoptotic process, 87% were homologs of *A. thaliana* NLR genes, a class of genes best known for its involvement in perception and response to pathogen attack (*Jones and Dangl, 2006*). An even higher enrichment for ts$_{CgCr}$SNPs was found when testing this class of genes specifically, with ts$_{CgCr}$SNPs falling in NLR genes being more likely than those in other types of genes to result in nonsynonymous changes

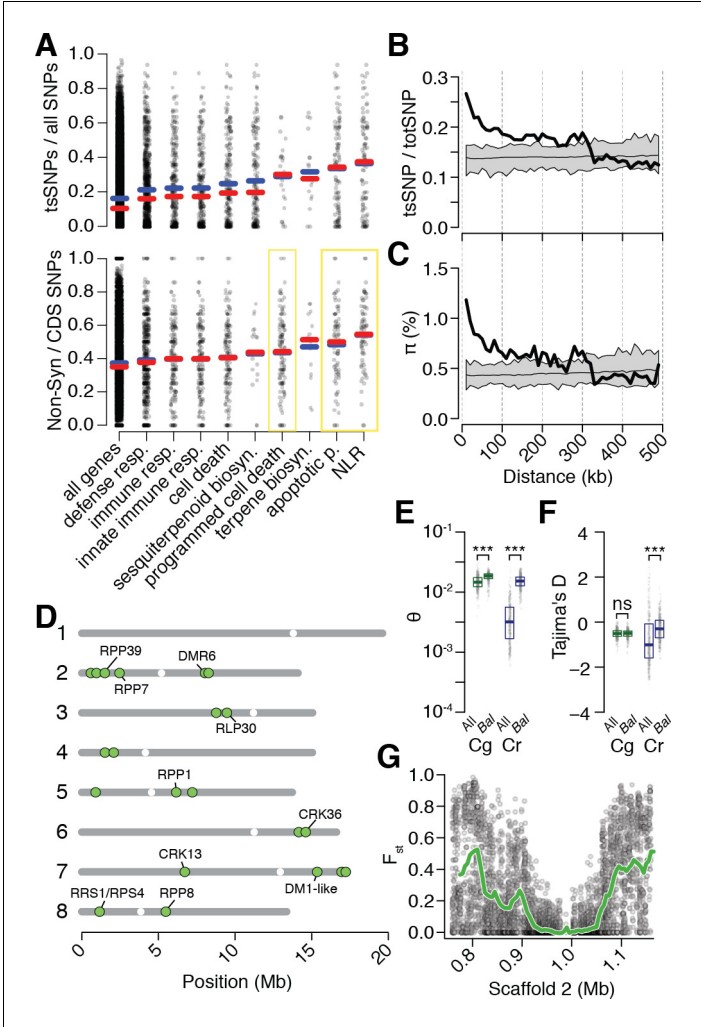

**Figure 4.** Preferential sharing of alleles near immunity genes. (A) Enrichment of $ts_{CgCr}$SNPs and non-synonymous $ts_{CgCr}$SNPs for genes associated with significant GO terms (means, blue; medians, red). Each point represents values calculated for an individual gene. For example, in the upper subplot each point is the number of tsSNPs identified in a gene divided by the total number of SNPs identified for a gene. GO terms with a significantly increased ratio of nonsynonymous changes are highlighted with a yellow box. (B) $ts_{CgCr}$SNP frequency as a function of distance to the closest NLR cluster. (C) Pairwise genetic diversity at neutral (four-fold degenerate) sites as a function of distance to the closest NLR cluster. For (B–C) the thick black lines are mean values calculated in 500 bp windows as a function of distance from a NLR gene. The thin black lines are mean values from 100 random gene sets of equivalent size. The grey polygons are the range of values across all 100 random gene sets. (D) Chromosomal locations of *Bal* regions with the strongest evidence for balancing selection. Immunity genes with known function in *A. thaliana* in each region indicated. (E) Values for Watterson's estimator ($\Theta_w$) of diversity in *Bal* regions, calculated from 20 kb windows. (F) Tajima's D. Dots in (E, F) are a random sample of 1000 windows for non candidate windows. Boxplots report the median 1st and 3rd quantiles of all windows in each class. (G) An example of the site level (dots) and windowed (green) decrease in $F_{st}$ at the first region on chromosome 2. The subregion without data near 1 Mb is a CC-NLR cluster, which was largely masked for variant calling.

DOI: https://doi.org/10.7554/eLife.43606.022

The following source data and figure supplements are available for figure 4:

**Source data 1.** Regions with evidence for balancing selection.

DOI: https://doi.org/10.7554/eLife.43606.025

**Source data 2.** Spearman's correlations of allele frequencies for different classes of tsSNPs.

DOI: https://doi.org/10.7554/eLife.43606.026

**Figure supplement 1.** Quality metrics for ssSNPs and tsSNPs in *C. rubella*.

DOI: https://doi.org/10.7554/eLife.43606.023

*Figure 4 continued*

**Figure supplement 2.** Analysis of IBS in balanced regions and genome-wide.

DOI: https://doi.org/10.7554/eLife.43606.024

(*Figure 4A–B*). These results indicate that despite a severe global loss of genetic diversity, genes involved in plant-pathogen interactions have maintained high levels of genetic variation in *C. rubella*.

While the high density of ts$_{CgCr}$SNPs near immunity genes was intriguing, NLR genes frequently occur in complex clusters, which could elevate error rates during SNP calling and thus potentially influence our analyses. Of particular concern is that sequencing reads derived from paralogs not found in the reference, but present in some accessions, could be mismapped against the reference, leading to false positive ts$_{CgCr}$SNPs calls. We therefore examined whether ts$_{CgCr}$SNPs showed more evidence of such errors than other SNPs. Mismapping should increase coverage and reduce concordance (the fraction of reads supporting a particular call) at a site. That the distributions of these two metrics were nearly identical at ts$_{CgCr}$SNPs and ssSNP sites indicates, however, that mismapping is unlikely to have affected our SNP calls (*Figure 4—figure supplement 1*). Mismapping is also expected to cause pseudo-heterozygous calls, due to reads from different positions in the focal genome being mapped to the same target in the reference genome. However, ts$_{CgCr}$SNPs were not more likely to be found in the heterozygous state as compared to ssSNPs (*Figure 4—figure supplement 1*). In addition, we asked whether the signal of increased ts$_{CgCr}$SNPs density extended into sequences adjacent to NLRs and is detectable even when masking the NLR clusters themselves. For this purpose, we collapsed NLR genes within 10 kb of one another into a single region, and calculated ts$_{CgCr}$SNPs rates and genetic diversity as a function of distance from these collapsed regions, ignoring SNPs within the focal cluster. We found that elevated ts$_{CgCr}$SNPs sharing and genetic diversity extended over 100 kb from NLR genes. Thus, increased sharing is not an artefact of the internal structure of NLR clusters (*Figure 4B–C*).

Increased retention of genetic diversity near immunity loci suggests that these genes might be the targets of balancing selection in either *C. rubella*, *C. grandiflora*, or both species. However, neutral processes including random introgression and stochastic allele fixation can give rise to uneven distributions of genetic variation across the genome after genetic bottlenecks (*Robinson et al., 2016*). We sought to identify regions that showed a pattern of allele sharing that was unlikely to have occured neutrally, as indicated by low values of the fixation index F$_{st}$, which quantifies genetic differentiation between populations. We compared the observed values of F$_{st}$ between *C. rubella* and *C. grandiflora* to a distribution calculated from simulated sequences under our previously inferred neutral demographic model, which included gene flow between *C. rubella* and *C. grandiflora*. We simulated one million 20 kb DNA segments, or just over 7,000 *C. rubella* genome equivalents, under the neutral model and calculated the expected distribution of F$_{st}$ values. Using this distribution, we assigned the probability of observing the F$_{st}$ value for each non-overlapping 20 kb window throughout the genome. After Bonferroni correction and joining of adjacent significant segments, we identified 21 genomic regions that we designated as candidate targets of balancing selection (*Bal*, *Figure 4D* and *Figure 4—source data 1*). *Bal* regions showed several classical indications of balancing selection including substantially higher Tajima's *D* and within-*C. Rubella* genetic diversity relative to the remainder of the genome (*Figure 4E–F*; p<<0.001 Mann-Whitney U-test for both statistics). ts$_{CgCr}$SNPs in *Bal* regions were also less likely to have been lost during colonisation of Western Europe than ss$_{Cr}$SNPs or ts$_{CgCr}$SNPs in other parts of the genome, and allele frequencies in *Bal* regions showed elevated correlation across populations (*Figure 4—source data 2*). Like tsSNPs in general, *Bal* regions did not show evidence for increased heterozygosity that might indicate increased error rates in SNP calling (Median Observed - Expected Heterozygosity in 20 kb windows was −0.021 inside of *Bal* regions, and −0.020 outside of these regions).

Estimates of F$_{st}$ were reduced in large segments surrounding NLR and other immune gene candidate clusters (*Figure 4G*), consistent with allele sharing being the result of linkage to a nearby balanced polymorphism. Of the 21 candidate regions, nine overlapped with clusters of NLR genes, and five with clusters of RLK/RLP or CRK genes, two classes of genes with broad roles in innate immunity (*Yeh et al., 2015*; *Zipfel, 2008*). Many of the specific regions we identified in *Capsella* have been directly demonstrated to function in disease resistance in *A. thaliana* (*McDowell et al., 2000*; *McDowell, 1998*; *Goritschnig et al., 2012*; *Holub, 1994*; *Gassmann et al., 1999*; *Zhang et al., 2014*;

*Zhang et al., 2013*; *Xu, 2006*; *Yeh et al., 2015*). *RPP1* and *RPP8* have been previously suggested as candidate targets of balancing selection, and trans-specific polymorphism has been reported at the *RPP8* locus in the genus *Arabidopsis* (*Bergelson et al., 2001*; *Wang et al., 2011*). It should be noted, however, that these genes are often members of larger linked NLR gene superclusters, with some of the regions our approach identified being sizeable and thus making it difficult to pinpoint a single focal gene. Indeed the strong signal found in these regions could result from multiple linked balanced sites. Furthermore, the strongest signals of balancing selection are mostly derived from linked sites, rather than the clusters themselves, because confident SNP calling is very difficult, if not impossible, with short reads in the most complex genomic regions (*Figure 4G*).

It is formally possible that the unusual pattern of diversity that we observe near *Bal* loci could result from historical balancing selection in the outcrossing ancestor *C. grandiflora* rather than ongoing selection in the selfing *C. rubella*. Population genetic indices such as $F_{st}$, nucleotide diversity pi, Tajima's D, and allele sharing are not fully independent, and elevated diversity in the *C. rubella* founding population, driven by historical balancing selection, could also generate the observed patterns. Genetic diversity was only modestly elevated in these regions in *C. grandiflora* (p<<0.001 Mann-Whitney U-test, *Figure 4E*), and Tajima's *D* was not significantly different from other windows (*Figure 4F*), suggesting that this is not very likely. If balancing selection is acting at these loci in the outcrosser, it is clear that its genomic footprint is small, perhaps due to the rapid decay of LD in this species relative to the selfing *C. rubella*. Still, it is possible that even a small elevation of genetic diversity in *Bal* regions in the founding populations might have considerable impact on subsequent *C. rubella* diversity. We approximated this situation using our simulated genetic data. We subsetted simulations by the level of genetic diversity in *C. grandiflora*, choosing the top 1% of simulated values. Even in the case of elevated founder diversity in these data, the observed $F_{st}$ values in *Bal* regions remain exceptionally unlikely (p<0.0001). These observations point to ongoing balancing selection within *C. rubella* maintaining diversity in *Bal* regions.

Adaptive retention of *C. grandiflora* diversity in *Bal* regions could be explained by two non-exclusive models. Allelic variation might have been present in the *C. rubella* founding population and maintained by balancing selection until the present. Alternatively, beneficial alleles may have been introgressed from *C. grandiflora* after the evolution of selfing, and retained by balancing selection. We searched for evidence of recent ancestry between the two species in *Bal* regions. A larger fraction of *Bal* region sequence was found to be IBD when compared to the genome-wide average (*Figure 4—figure supplement 2*), consistent with elevated retention of introgressed alleles in these regions. Shared segments in *Bal* regions were on average shorter than those found in other parts of the genome, suggesting that they are older and have been subjected to longer periods of recombination since the introgression event (median within 3,503 bp, median outside 6,661 bp, Wilcoxon-rank sum test, p=1e-54), although we cannot exclude the influence of differing patterns of recombination in these regions as a contributing factor to this observation.

Elevated IBD rates in *Bal* regions might result from gene flow between the species in either direction, and our previous results suggested that most modern gene flow occurs through introgression of *C. rubella* alleles into *C. grandiflora*. We explored the geographic pattern of IBD between *C. rubella* and *C. grandiflora* in *Bal* regions to determine whether it differs from that of the genome-wide pattern. Within the East population, IBD decayed as a function of distance from the *C. grandiflora* range in a manner comparable to the observed genome-wide pattern, albeit with a more shallow slope (*Figure 4—figure supplement 2*). In contrast to the genome-wide pattern, high levels of IBD were observed between *C. grandiflora* and West population accessions. Thus, we find evidence for neutral gene flow throughout the genome, perhaps dominated by *C. rubella* to *C. grandiflora* introgression, as indicated by our demographic simulations. However, allele sharing appears to be older in *Bal* regions and introgressed alleles have been retained for longer periods even after colonisation of Western Europe. This latter observation is consistent with the hypothesis that alleles were introgressed prior to the most recent range expansions in *C. rubella*, and that variation was subsequently maintained by selection in *Bal* regions.

## Balancing selection over millions of years

Although evidence for balancing selection at immunity-related loci in *C. rubella* is much stronger than in *C. grandiflora*, it is difficult to completely exclude the effect of founder diversity at these loci on the observed patterns. We therefore sought to validate our findings in a related species that has

been separated from *C. grandiflora* and *C. rubella* for a long time. The genus *Capsella* offers a unique opportunity to test the longevity of balancing selection, because selfing has evolved independently in *C. orientalis*, which diverged from *C. grandiflora* and *C. rubella* more than one million years ago and whose modern range no longer overlaps with the two other species, preventing ongoing introgression (*Hurka et al., 2012*; *Douglas et al., 2015*). We expected the evolution of selfing to have generated a similar bottleneck as in *C. rubella* (*Douglas et al., 2015*; *Bachmann et al., 2018*), and we therefore resequenced 16 *C. orientalis* genomes, to test whether there is evidence of balancing selection at similar types of loci.

After alignment, SNP calling, and filtering, we identified a mere 71,454 segregating SNPs in *C. orientalis*. This is a surprisingly small amount of variation, corresponding to an almost 50-fold reduction in diversity relative to the outcrossing *C. grandiflora* (*Figure 5—source data 1*). Using our divergence and diversity measures, we estimated that *C. orientalis* diverged from *C. grandiflora* over 1.8 million generations ago (calculated as in ref. *Brandvain et al., 2013*). The combination of long divergence times and low variability in *C. orientalis* makes it unlikely that alleles will have been maintained by random chance. Using estimates of $N_e$ from nucleotide diversity at four-fold degenerate sites (*C. orientalis* [14,643] and *C. grandiflora* [694,643]), the divergence time above, and the genome assembly size of 134.8 Mb, the probability of finding a single tsSNP is $<4 \times 10^{-19}$ using the methodology of Leffler and colleagues and Wiuf and colleagues (*Leffler et al., 2013a*; *Wiuf et al., 2004*), which assumes constant population size. It was therefore surprising that 8,408 *C. orientalis* variants were shared with either *C. rubella* or *C. grandiflora* (ts$_{2\text{-way}}$SNPs), and 3992 with both (ts$_{3\text{-way}}$SNPs, *Figure 5A–B*). In each of the three species, ts$_{3\text{-way}}$SNPs were enriched at higher derived allele frequencies relative to ssSNPs and ts$_{2\text{-way}}$SNPs, suggesting that they are on average the oldest SNPs (*Figure 5C*).

Because this large amount of trans-specific polymorphism was unexpected, we wanted to ensure that this was not due to more error-prone read mapping to a distant reference. We therefore also used an additional set of more stringent filters to identify high confidence ts$_{3\text{-way}}$SNPs (ts$_{3\text{-wayhq}}$SNPs; see Materials and methods). Importantly, we required ts$_{3\text{-wayhq}}$SNPs to be in LD with at least one other ts$_{3\text{-wayhq}}$SNP in all three species ($r^2 > 0.2$ in the same phase), to provide evidence that they represented the same ancestral haplotype. The aim was to improve the likelihood that such SNPs were true examples of identity by descent. Furthermore, we generated a draft assembly of the *C. orientalis* genome using Pacific Biosciences SMRT cell technology, and re-called ts$_{3\text{-way}}$SNP sites. We identified 812 high quality transpecific SNPs segregating in all three species (ts$_{3\text{-wayhq}}$SNPs). The distributions of coverage and concordance values in this dataset were similar between ts$_{3\text{-way}}$SNP sites and other *C. orientalis* sites, further supporting their authenticity (*Figure 5—figure supplement 1*).

As discussed earlier, the presence of trans-specific polymorphism in diverged species could be driven by stable balancing selection or it could result from gene flow between the species. While *C. grandiflora* and *C. rubella* occur around the Mediterranean, *C. orientalis* is restricted to Central Asia (*Hurka et al., 2012*) and its current distribution is far from that of *C. grandiflora* and *C. rubella*. Modern gene flow between the *C. orientalis* and *C. rubella*/*C. grandiflora* lineages is therefore unlikely, but it is possible that the ranges of these species overlapped in the past. If alleles have been maintained since the split between the lineages, then the divergence between maintained alleles should meet or exceed the divergence between the species. On the other hand, if ts$_{3\text{-wayhq}}$SNPs are the result of recent gene flow between the lineages, then divergence between species near these SNPs should be reduced compared to the genome-wide average divergence. We examined diversity and divergence at neutral (four-fold degenerate) sites surrounding ts$_{3\text{-wayhq}}$SNPs (*Figure 5D*). In all three species, diversity was high directly adjacent to ts$_{3\text{-wayhq}}$SNPs, close to average levels for genome-wide divergence between the two *Capsella* lineages. This footprint of elevated diversity is much more discernible in the two selfing species than in *C. grandiflora*. No obvious reduction in divergence was observed near ts$_{3\text{-wayhq}}$SNPs (*Figure 5D*). We conclude that ts$_{3\text{-wayhq}}$SNPs correspond predominantly to long-term maintained alleles that diverged on ancient time scales and that they are not the result of recent introgression.

The finding of tsSNPs shared between two independent lineages, *C. grandiflora*/*C. rubella* and *C. orientalis*, for over a million generations in spite of strong geographic barriers suggests that they are targets of stable long-term balancing selection. If this selection pressure remains constant across species, ancient alleles are expected to evolve towards similar equilibrium intermediate frequencies.

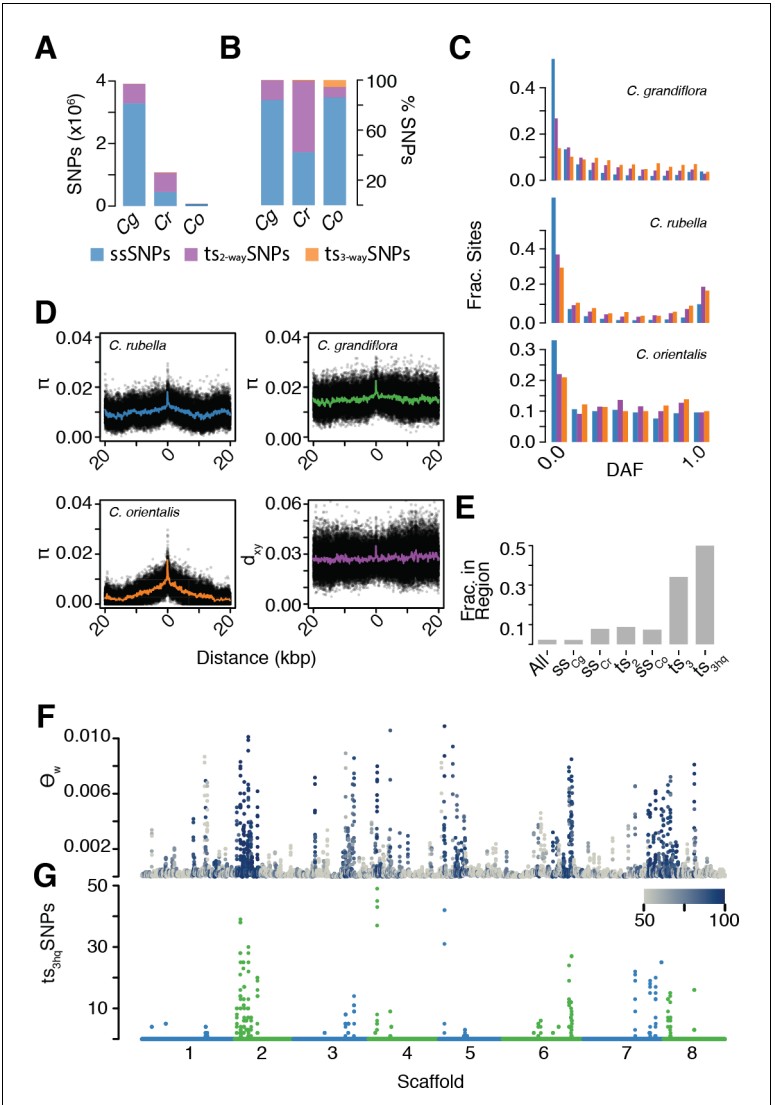

**Figure 5.** The signal of ancient balancing selection. (**A**) Absolute number and (**B**) fraction of ssSNPs (blue), ts$_{2\text{-}way}$SNPs (purple), and ts$_{3\text{-}way}$SNPs (orange) for *Capsella*. Only sites accessible to read mapping in all three species were considered. (**C**) Derived allele frequency (using *A. lyrata* and *A. thaliana* as outgroup) of ssSNPs (blue), ts$_{2\text{-}way}$SNPs (purple) and ts$_{3\text{-}way}$SNPs (orange). (**D**) Pairwise diversity (π) as a function of distance from a ts$_{3\text{-}way}$SNP in *C. grandiflora*, *C. rubella*, and *C. orientalis*. Bottom right, Divergence between the *C. rubella*/*C.grandiflora* and *C. orientalis* lineages as a function of distance from a ts$_{3\text{-}way}$SNP. Black dots, means over all sites at a particular distance, and coloured lines, means over bins of 50 bp. (**E**) Enrichment of ts$_{3\text{-}way}$SNPs and ts$_{3\text{-}wayhq}$SNPs in candidate balanced regions from the *C. rubella*/*C. grandiflora* comparison. (**F**) Watterson's estimator (Θ$_w$) of genetic diversity in *C. orientalis*, in 20 kb windows. Grey-to-blue scale indicates the genome-wide percentile of the same window for Θ$_w$) in *C. rubella*. (**G**) ts$_{3hq}$SNP number in each window.

DOI: https://doi.org/10.7554/eLife.43606.027

The following source data and figure supplements are available for figure 5:

**Source data 1.** Three species diversity and divergence.
DOI: https://doi.org/10.7554/eLife.43606.030
**Figure supplement 1.** Distributions of concordance and coverage values for different SNP classes in *C. orientalis*.
DOI: https://doi.org/10.7554/eLife.43606.028
**Figure supplement 2.** tr$_{3\text{-}wayhq}$SNPs MAF.
DOI: https://doi.org/10.7554/eLife.43606.029

In comparison to ts$_{2\text{-way}}$SNPs, the minor alleles at ts$_{3\text{-wayhq}}$SNP sites are closer to intermediate frequencies in all three species (*Figure 5—figure supplement 2*). Furthermore, ts$_{3\text{-wayhq}}$SNPs segregate at more similar allele frequencies in *C. rubella* and *C. grandiflora* than other two-way tsSNPs, as measured by F$_{st}$ median values: 0.03 for ts$_{3\text{-wayhq}}$SNPs and 0.16 for ts$_{2\text{-way}}$SNPs, p<<0.001 Mann-Whitney test) and correlation of derived allele frequencies (*Figure 4—source data 2*). These results suggest a conserved equilibrium maintained since the isolation of *C. rubella* and *C. grandiflora* over 10,000 generations ago. Derived allele frequencies for ts$_{3\text{-wayhq}}$SNPs are not correlated between the two ancient *Capsella* lineages (Spearman's rho −0.08 *C. orientalis* to *C. grandiflora* and −0.04 to *C. rubella*). It is possible that demographic reduction or habitat shift in *C. orientalis* has disturbed this equilibrium.

Like ts$_{CgCr}$SNPs, ts$_{3\text{-way}}$SNPs are strongly enriched in GO categories associated with immunity (*Supplementary file 2*). Our previously identified balanced regions strongly predicted the genomic distribution of ts$_{3\text{-way}}$SNPs; 50% of ts$_{3\text{-wayhq}}$SNPs fell into these regions, even though they encompass fewer than 10% of ts$_{CgCr}$SNPs and fewer than 3% of all SNPs, resulting in an even more skewed and uneven distribution of genetic diversity along the genome (*Figure 5F–G*). At least one ts$_{3\text{-wayhq}}$SNP was found in each of 10 of the 21 original candidate regions under balanced selection. Six of these corresponded to NLR clusters, two to RLK/RLP clusters, and one to a TIR-X cluster. Only one region did not contain a clear immunity candidate, with the caveat that this conclusion is based on the single annotated *C. rubella* reference genome (*Slotte et al., 2013*). Thus, even in a situation where a recent genetic bottleneck has wiped out almost all genetic diversity, there is very strong selection to maintain allelic diversity at specific immunity-related loci, consistent with these alleles having persisted already for very long evolutionary times.

## Insights into balancing selection from de novo assembly of *MLO2*

The balanced regions we identified contained very old tsSNPs, yet as mentioned, the immunity genes themselves are often not accessible to variant discovery based on mapping short reads to a single reference genome. Furthermore, it is possible, or even likely, that the strongest evidence for balancing selection comes from loci that include several linked targets of balancing selection. This combination of factors makes it difficult to pinpoint potential functional changes maintained by balancing selection in these regions. To discover functional changes, we therefore focused on ts$_{3\text{-wayhq}}$SNPs that did not fall in our large balanced regions but were clustered in regions of the genome that were likely less complex. We selected genes that were well covered by reads in all three species (>80% sites), contained at least six high quality tsSNPs, at least one non-synonymous ts$_{3\text{-wayhq}}$SNP, were at least 100 kb from any of our candidate balanced regions, and had been functionally characterised in *A. thaliana*. These filters singled out a homolog of the *A. thaliana MLO2* gene as a particularly good candidate for more detailed analysis (*Supplementary file 3* and *Figure 6*).

*MLO2* encodes a seven-transmembrane domain protein with a conserved role in plant disease susceptibility (*Figure 6A*) (*Consonni et al., 2006*). The *C. rubella MLO2* locus has experienced a tandem duplication, resulting in two genes, *MLO2a* and *MLO2b*. Although both homologs are sufficiently diverged to be accessible to unambiguous read mapping, all six ts$_{3\text{-wayhq}}$SNPs were in *MLO2b* (*Figure 6B–C*). In *C. rubella* and *C. orientalis*, the ts$_{3\text{-wayhq}}$SNPs were arranged in five different haplotypes, which we collapsed into three related haplogroups, A, B and C (*Figure 6B*). The reference haplogroup A was most frequent in both species.

Because several known targets of balancing selection in *A. thaliana* are the result of structural variation, or lesions larger than 1 kb (*Mauricio et al., 2003*; *Stahl et al., 1999*), we examined coverage patterns around the *MLO2* locus to identify potential linked indels. We found that haplogroup B in both *C. rubella* and *C. orientalis* exhibited similar patterns of low read coverage at the 5′ end of *MLO2b*, suggesting a possible indel (*Figure 6C*). To examine the exact sequence of each allele, we took advantage of the homozygous nature of sequence data from these two selfing species and performed local de novo assembly of the *MLO2* locus from read pairs mapping to this region. We were able to reconstruct the locus for 15 *C. orientalis* samples (13 haplogroup A and two haplogroup B) and 43 *C. rubella* samples (34 A, 2 B, and 7 C). Surprisingly, a comparison of the different haplotypes revealed that the pattern of low coverage observed for haplogroup B was not due to structural variation, but instead to extremely high divergence from the reference haplogroup A (*Figure 6—figure supplement 1*). Divergence between alleles within species was greater than 0.15 differences per bp,

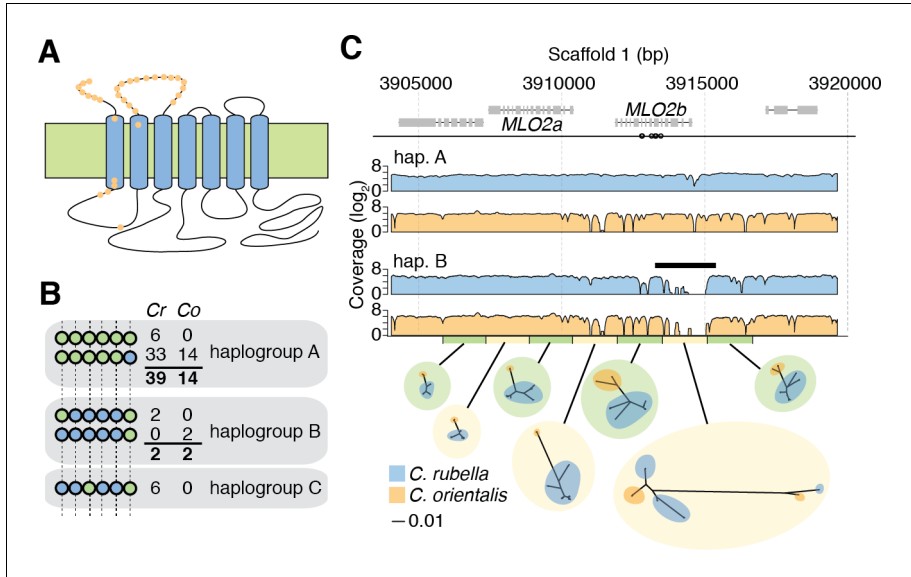

**Figure 6.** Evidence for long-term balancing selection at *MLO2*. (**A**) Diagram of MLO2 protein in the cell membrane. Blue ovals, transmembrane domains. Top is the extracellular space. Orange dots represent amino acid differences between proteins encoded by haplogroups A and B. (**B**) Haplogroup identification with reference based SNP calls. Circles represent ts$_{3-wayhq}$SNPs and colours represent the reference (green) and alternative (blue) SNP calls. Numbers indicate haplotype frequencies in each species. (**C**) Top: A diagram of the *MLO2* region on scaffold 1. Grey boxes represent coding regions. Empty circles show the positions of the seven initially identified ts$_{3-wayhq}$SNPs shown in (**B**). *MLO2b* gene is drawn based on the reference annotation, but alignment with orthologous genes suggested a misannotation of the last splice site acceptor leading to truncation of the annotated gene. For final alignments, the corrected annotation was used. Middle: Average read coverage by haplogroup and species (blue is *C. rubella* and orange is *C. orientalis*). Region of poor coverage in haplogroup B is highlighted with a black bar. The green and yellow bars below the coverage plots highlight the de novo assembled region and the windows from which the neighbour-joining trees were built (excluding indels, each window is 1 kb). The blue and orange circles on the tree indicate samples from each species. Black scale indicates substitutions in trees.

DOI: https://doi.org/10.7554/eLife.43606.031

The following figure supplements are available for figure 6:

**Figure supplement 1.** Sliding windows of allelic divergence and positions of tsSNPs.
DOI: https://doi.org/10.7554/eLife.43606.032
**Figure supplement 2.** Phylogenetic analysis of the MLO2 CDS.
DOI: https://doi.org/10.7554/eLife.43606.033
**Figure supplement 3.** Alignment of amino acid sequences at the MLO2 N-terminus.
DOI: https://doi.org/10.7554/eLife.43606.034

over three times higher than the genome-wide divergence between the species (*Figure 6—figure supplement 1* and *Figure 5—source data 1*). This highly diverged region had therefore been originally inaccessible to reference-based read mapping in haplogroup B samples. De novo assembly allowed us to identify a total of 204 additional tsSNPs, nearly all of which mapped to the 5' end of *MLO2b* (*Figure 6—figure supplement 1*). Neighbour-joining trees revealed the expected clustering of samples by species in regions adjacent to *MLO2b*, but clear clustering by haplogroup within the 5' region, a pattern that is reproduced in phylogenetic analysis of the entire CDS (*Figure 6C* and *Figure 6—figure supplement 2*). Importantly, divergence within haplogroup across species was greater than, or similar to genome-wide averages for both A and B, demonstrating that recent introgression did not give rise to allele sharing (*Figure 6—figure supplement 1*).

The high nucleotide divergence between haplogroups A and B translates into numerous amino acid differences in the N terminal half of the encoded proteins. In a 157 amino acid stretch, 31 amino acid differences are found in both species (*Figure 6A* and *Figure 6—figure supplement 3*), with an indel polymorphism accounting for another seven amino acid differences. The large number of

differences between the two haplogroups makes it difficult to point to any specific change as the target of balancing selection, but it seems likely that the two alleles differ functionally, perhaps reinforced by additional differences in the promoter. In summary, the nucleotide divergence in this region suggests that the *MLO2b* haplogroups are much older than the split between the two species.

## Discussion

While balancing selection has long been recognised as an important evolutionary force, its relevance as a major factor shaping genomic variation has remained unclear (*Charlesworth, 2006*; *Wiuf et al., 2004*; *Asthana et al., 2005*). We have taken advantage of unique demographic situations in two *Capsella* lineages to demonstrate not only that there is pervasive balancing selection at immunity-related loci in this genus, but also that the same alleles are maintained in species that are likely experiencing quite different pathogen pressures. We expect that balancing selection plays a similar role in other taxa, but that its effects are masked by a background of higher neutral genetic diversity and more frequent recombination between balanced sites and linked variants (*Wiuf et al., 2004*; *Charlesworth, 2006*). In addition, the detection of long-term balancing selection is further compounded by very old alleles being less accessible to short read re-sequencing, the dominant mode of variant discovery today. In the two selfing *Capsella* species, the footprints of balancing selection extend for tens of kilobases, greatly impacting diversity of many other genes. While this makes it more difficult to pinpoint the actual selected variants, it greatly improves statistical power to identify regions under balancing selection. This is reminiscent of genome-wide association studies, where extended LD improves statistical power to detect causal regions of the genome but reduces the ability to identify the specific causal variants (*Atwell et al., 2010*).

The nature of balancing selection acting on the regions we have identified remains to be clarified. Stable balancing selection in self fertilising species is unlikely to derive from heterozygous advantage, pointing to negative frequency-dependent selection or fluctuating selection from variable pathogen pressures as possible factors. While the mode of selection cannot be determined from these static data, the strong signal that we observe in highly selfing lineages points to environmental heterogeneity or negative frequency dependent selection over heterozygote advantage. Based on the enrichment of immunity-related genes, it appears that biotic factors are the dominant drivers of long-term maintenance of polymorphism. This observation is consistent with a large body of work on intraspecific variation in *A. thaliana*. The signal of balancing selection has been observed for specific pairs of disease resistance alleles in *A. thaliana* (*Stahl et al., 1999*; *Tian et al., 2002*; *Tian et al., 2003*; *Mauricio et al., 2003*; *Bakker, 2006*), and in the case of the resistance gene *RPS5*, alternative alleles have been shown to affect fitness in the field (*Karasov et al., 2014*). It is possible, or perhaps even likely, that the signal of balancing selection is amplified by the fact that immunity-related loci occur in clusters (*Meyers, 2003*) and that our strongest signal is the result of simultaneous selection on several genes in these regions in a situation analogous to the MHC in animals (*Hedrick, 1998*). Thus, biotic factors might not be quite as important as our analyses make them appear. On the other hand, it is also possible that the clustering of disease resistance genes itself is a product of selection, if selection was more effective when acting on groups of genes (*Charlesworth and Charlesworth, 1975*), or if evolution under a balanced regime was deleterious at other types of loci. Even if we accept that biotic factors predominate, the nature of the potential trade-offs that prevent individual alleles from becoming fixed is still a mystery, but it might involve conflicts between growth and defense (*Coley et al., 1985*; *Walling, 2009*; *Herms and Mattson, 1992*), beneficial and harmful microbe interactions (*Walters and Heil, 2007*), or defense against different types of pathogens (*Kliebenstein and Rowe, 2008*). What is clear is that the trade-offs must be stable over very long periods of evolution.

Our findings suggest a model in which the success of self fertilising populations may be buoyed by gene flow from outcrossing relatives in a situation analogous to evolutionary rescue strategies in conservation biology (*Whiteley et al., 2015*). This model is a variation on the theme of adaptive introgressions, which have recently emerged as a major evolutionary force in a wide range of taxa (*Whitney et al., 2006*; *Castric et al., 2008*; *Pease et al., 2016*; *Dasmahapatra et al., 2012*; *Henning and Meyer, 2014*; *Hedrick, 2013*; *Huerta-Sánchez et al., 2014*; *Racimo et al., 2015*; *Castric et al., 2008*; *Pease et al., 2016*; *Dasmahapatra et al., 2012*; *Whitney et al., 2006*;

*Huerta-Sánchez et al., 2014*; *Hedrick, 2013*). The unique feature of self-fertilisation in comparison to these examples is that the amplified effects of linked selection and genetic drift lead to a steady loss of genetic variation over time. Constant replenishment via adaptive introgression from an outcrossing relative counters the loss of diversity at immunity-related loci, thereby preventing decreased fitness in competition with pathogens. Whether this model generally applies will require independent study of other lineages of related self-fertilising and outcrossing populations at various stages of speciation.

Finally, we note that maintenance of ancient variants is most easily detectable in a background of low variation. Therefore, it could potentially be used to rapidly identify loci with meaningful functional variation. Typically, agricultural breeding panels seek to maximise surveyed diversity, but our results indicate that identification of useful immunity-related polymorphism with genomic data might be facilitated in otherwise homogeneous wild populations.

## Materials and methods

### Plant material and DNA extraction

Seeds were stratified for two weeks at 4°C and germinated in controlled environment chambers. Four to six rosette leaves were collected from each accession and frozen in liquid nitrogen for gDNA extraction. The methods available for extraction and sequencing varied as the project progressed, and 24 of the *C. rubella* and the 13 *C. grandiflora* samples were analysed independently in previous studies (*Agren et al., 2014*; *Williamson et al., 2014*). See *Figure 1—source data 1* for a listing of DNA preparation, library construction, and sequencing technology by sample. In brief, DNA was extracted following an abbreviated nuclei enrichment protocol (*Becker et al., 2011*) or using the Qiagen Plant DNeasy Extraction kit. The recovered DNA was sheared to the desired length using a Covaris S220 instrument, and Illumina sequencing libraries were prepared using the NEBNext DNA Sample Prep Reagent Set 1 (New England Biolabs) or the Illumina TruSeq DNA Library Preparation Kit and sequenced on the instrument as listed in *Figure 1—source data 1*. We aimed for a minimum genome coverage of 40x. We mapped reads to the *C. rubella* reference genome (*Slotte et al., 2013*) resulting in realised coverages of 30 – 126x.

### Sequence handling and variant calling

Initial sequence read processing, alignment, and variant calling were carried out using the SHORE (v0.8) software package (*Ossowski et al., 2008*). Read filtering, de-multiplexing, and trimming were accomplished using the import command discarding reads that had low complexity, contained more than 10% ambiguous bases, or were shorter than 75 bp after trimming. Reads were mapped to the *C. rubella* reference genome (Phytozome v.1.0) using the GenomeMapper aligner (*Schneeberger et al., 2009*) with a maximum edit distance (gaps or mismatches) of 10%. Alignments from each sample were then processed to generate raw whole genome reference and variant calls with qualities computed using an empirical scoring matrix approach (*Cao et al., 2011*) allowing heterozygous positions. Of the initial 53 *C. rubella* samples, two were removed because of low or uneven coverage, and one was removed as a misidentified *C. bursa-pastoris* sample (*C. rubella* and its polyploid relative *C. bursa-pastoris* are not easily identified phenotypically, but they can be distinguished by the extreme number of pseudo-heterozygous calls in the latter).

The per-sample raw consensus calls produced by SHORE were used to construct a whole genome matrix of finalised genotype calls for each species. Positions were considered only if covered by at least four reads and if overlapping reads mapped uniquely (GenomeMapper applies a 'best match' approach, so unique means that only one best match exists) (*Schneeberger et al., 2009*). We simultaneously considered information from all samples within a species to make base pair calls. If no variant was called in any sample then the site was treated as reference. Individual sample calls were made if four reads supported the reference base, the computed quality was above 24, and at least 80% of reads supported a reference call. A site was excluded if more than 30% of the samples from that species did not meet these criteria.

If at least one sample reported a difference from the reference in the raw consensus, then variant (indel or SNP) or reference calls were considered. The SNP calling parameters were slightly different for the two selfing species as compared to the outcrossing *C. grandiflora* because variants should

only rarely be found in the heterozygous state in the former (and the frequency of heterozygous calls in a selfing species is a powerful filter to detect problems with mismapped reads). The general approach was to require at least one high quality variant call at a site and then to call genotypes in other samples with slightly reduced stringency. If no variant call met the more stringent threshold, then the site was reconsidered using the above reference criteria. Finally, the calls from each of the three species were combined into a master matrix. If a position was not called biallelic or invariant across the compared species, then it was not considered. To facilitate further analyses in PLINK (v1.9) (*Chang et al., 2015*) and vcftools (v0.1.12a) (*Danecek et al., 2011*), the genome matrix at biallelic SNP sites was also converted into a minimal vcf format.

## Defining pericentromeric regions

Regions of high repeat density near the centromeres of all chromosomes as well as two large, repeat-rich regions in chromosomes 1 and 7 were removed from genome scans. Coordinates for these regions are listed in *Supplementary file 4*.

## Site annotations

We used the SnpEff (v.3.2a) (*Cingolani et al., 2012*) software package to annotate variant and invariant sites for the whole genome. The annotation database was built using the *C. rubella* v1.0 Phytozome gff file. Sites were annotated using the table input function that includes annotation of fold degeneracy for each site in coding regions. Invariant sites were annotated using a table with dummy SNPs at each position. The SnpEff program outputs several annotations for some sites, and a primary annotation was selected by ranking the strength of effect of each annotation and reporting the annotation with the strongest effect (the rankings are listed in *Supplementary file 5*).

## Ancestral state assignment

To calculate derived allele frequency spectra we assigned ancestral state to each polymorphic site using three-way whole genome alignments between *C. rubella*, *A. thaliana*, and *A. lyrata* (*Slotte et al., 2013*). Only biallelic sites identical between *A. lyrata* and *A. thaliana* (indels were ignored) were considered. For the two species analysis, only sites also fixed for the ancestral allele in *C. orientalis* were considered.

## Trans-specific SNP annotation comparisons

To compare tsSNP and ssSNP annotations from similar allele frequency spectra, we binned 20,000 tsSNPs randomly drawn from throughout the genome by derived allele frequency (10 bins). We then drew an equivalent number of ssSNPs from each allele frequency bin and calculated the fraction of CDS SNPs that caused nonsynonymous changes and the fraction that fell in genes. This process was repeated 1000 times for both species to generate the plots shown in *Figure 3B*.

## Analysis of population structure and demographic modeling

Genotypes at four-fold degenerate SNP sites called in *C. grandiflora* and *C. rubella* were pruned in PLINK (50 kb windows, 5 kb step, and 0.2 r2 LD threshold) and used as input for ADMIXTURE (v.1.23) (*Alexander et al., 2009*) and EIGENSTRAT (v6.0 beta) (*Price et al., 2006*). For demographic modelling in Fastsimcoal (v2.5.2.11) (*Excoffier et al., 2013*), joint minor allele frequency spectra were generated at four-fold degenerate sites with complete information and ignoring heterozygous calls in selfing lineages (counting only one allele from each individual). Demographic parameters for each tested model were then inferred in 50 runs of Fastsimcoal (parameters: -l40 -L40 -n100000 -N100000 -M0.001 -C5). The global maximum likelihood model was selected after correcting for number of estimated parameters using Akaike Information Criterion. Confidence intervals were set for estimated parameters using 100 bootstraps of identical inference runs on simulated data under the most likely model. To reduce computational times, global maximum likelihoods were calculated for bootstraps after 13 runs rather than 50. The mutation rate assumed for this and other analyses was $7 \times 10^{-9}$ mutations/generation/ bp based on mutation rate measurements in *Arabidopsis thaliana* (*Ossowski et al., 2010*).

## Segments of recent ancestry and interspecific introgression

Segments of IBD were identified using the phasing and segment identification in Beagle (r1339) (*Browning and Browning, 2013*). For the analysis presented here, we considered only the first haplotype from each *C. rubella* sample and both haplotypes from each *C. grandiflora* sample. Segments were required to be larger than 1 kb to be considered in the analyses. D statistics were calculated as in *Green et al. (2010)*; *Patterson et al. (2012)*; *Dasmahapatra et al. (2012)* comparing each individual genotype from the eastern *C. rubella* population to allele frequencies from western *C. rubella* and *C. grandiflora*. The outgroup species for these analyses was *C. orientalis*.

## Sliding window analysis of genetic diversity

Population genetic diversity statistics for genome scans were calculated for each species by transforming variant calls from the genome matrix into FASTA files and inputting these files into the compute function from the libsequence analysis package (*Thornton, 2003*). Heterozygous bases were randomly assigned as reference or variant to generate a single haplotype for each sample. Weir and Cockerham's $F_{st}$ was calculated using vcftools (v.0.1.12a) on biallelic SNP sites.

## Identification of balanced regions

To identify regions of the genome with unusually low $F_{st}$ after speciation, we generated a null distribution of $F_{st}$ values by simulating one million 20 kb segments under our inferred best demographic model using Fastsimcoal2. The output of each simulation was transformed to vcf format and $F_{st}$ between *C. grandiflora* and each *C. rubella* subpopulation was calculated using vcftools. The probability of a particular $F_{st}$ value in the observed data was then assigned based on its rank in these simulations (independently for the two subpopulations; one sided test). Multiple testing was accounted for using Bonferroni correction. Significant outlier windows (adjusted p-value<0.05) identified for each subpopulation were collapsed into regions using a two state hidden markov-model as implemented in the Rhmm package. The HMM approach has the advantage of joining windows of high coverage separated by a low coverage window. Only regions significant in both subpopulations were considered for further analysis. Windows overlapping the pericentromeric regions were removed from the analysis.

## Linkage disequilibrium

LD was calculated in 30 kb windows in *C. grandiflora* and *C. rubella* using PLINK (v.1.9). The decay of LD is the mean value at each position up to 30 kb from a focal SNP.

## Gene ontology (GO) enrichment

Because the *C. rubella* annotation is sparse, we used annotations from nucleotide blast best hit matches (e < 1e-10) to CDS sequences from its close relative, *A. thaliana,* for our GO analysis. Enrichment tests were performed with the SNP2GO R library (*Szkiba et al., 2014*) using $ts_{CgCr}$SNPs as the test set and all SNP sites called in either *C. rubella* or *C. grandiflora* as the background set. We chose this approach because it is less sensitive to gene length (which should similarly affect tsSNP and non-tsSNP distributions across genes). A corresponding analysis was performed in the three-way comparisons using a background set of all SNP sites called in all three species. Significant enrichments were considered at a q-value threshold of q < 0.01 after false discovery correction. A gene was considered as belonging to the NLR family in *C. rubella* if its best blast hit in *A. thaliana* was annotated as such (*Supplementary file 6*).

## Identification of high quality three-way tsSNPs

To generate a list of high quality $ts_{3\text{-way}}$SNPs, we applied a series of empirical filters. First, all $ts_{3\text{-way}}$SNPs were required to have an $r^2 > 0.2$ with another $ts_{3\text{-wayhq}}$SNP in the same phase in all three species. We excluded SNPs overlapping pericentromeric or annotated repeat sequences (*Slotte et al., 2013*). We also required that the coverage of SNPs was no more than two standard deviations above the mean coverage of all SNPs for that species, to have an average concordance greater than 0.98, and to be identified in more than one individual. These criteria were selected to increase our confidence in identified tsSNPs; it is likely that our inferences are conservative.

To validate our trans-specific SNPs we aligned the *C. orientalis* samples against the draft *C. orientalis* assembly using the bwa (v.0.7.12) mem command with default parameters. The output bam format file was sorted using samtools (v.1.6) and multisample variant calls were made with freebayes (v.1.1.0) using the parameter settings -z. 1–0 w. The resulting vcf file was filtered using vcftools (v.0.1.13) using the settings –remove-indels –minQ 50 –max-missing 0.8 –max-alleles two and further filtered to remove sites that were called as heterozygous in more than 5% of the samples. The sites overlapping with the original call set were extracted from this vcf and used for validation.

Coordinate transforms between the two genomes were necessary to validate tsSNPs. The draft assembly of *C. orientalis* and the *C. rubella* reference genome were aligned using the LAST (v.923) aligner. The *C. rubella* reference database was built with the lastdb command with the parameter settings -uMAM8 -cR11, and then the two genomes were aligned with the lastal command with the settings -m50 -E0.05. Equivalent sites were considered if they were present in alignments at least 500 bp long and contained only one *C. orientalis* and one *C. rubella* sequence.

## Local *de novo* assembly and analysis of *MLO2*

To reconstruct alleles from the *MLO2* locus, we used an iterative assembly approach. Reads were first mapped to the entire reference genome using bwa (v.0.7.8) (*Li and Durbin, 2009*) using the bwa-mem alignment algorithm for each sample. Reads that mapped to the *MLO2* locus were then extracted and assembled de novo using SPAdes (v.3.5.0) (*Nurk et al., 2013*). Assemblies were filtered to be longer than 2,000 bp with a coverage greater than 5, and then used to create an index for a second round of read mapping. Reads that mapped to the assembly without mismatches were collected together with their mates (regardless of the mate's mapping quality), and were again de novo assembled. This process was iterated six times until scaffolds covering both coding regions were achieved. Format conversions and file handling made use of the software samtools (v.0.1.19) (*Li et al., 2009*) and bamutil (v.1.0.13).

Assemblies were filtered for appropriate length, and aligned using MAFFT (*Katoh and Standley, 2013*). Alignments were visualised using AliView (*Larsson, 2014*), and manually edited where appropriate. The protein encoded by *MLO2b* annotated in the *C. rubella* reference was truncated relative to *A. thaliana MLO2*. We aligned the genomic and coding regions from both species and found that the premature stop in *MLO2b* is likely due to a mis-annotated splice junction. The *A. thaliana* junction is conserved in *C. rubella* and alternative annotations on phytozome identify the *A. thaliana*-like splice variant. We therefore used the full-length version derived from manual alignments for our analysis. The phylogeny of *Capsella MLO2* CDS sequences was produced using the optim.pml command from the R package phangorn using Jukes-Cantor distances. 1000 bootstrap iterations were run to estimate support for nodes in the tree. To determine where amino acid substitutions had occurred, we aligned the proteins encoded by each allele against the barley *mlo* protein and annotated domains (UniProtKB P3766).

## Draft assembly of the *C. orientalis* genome

The draft genome from the *C. orientalis* accession 2007–03 (*Figure 1—source data 1*) was assembled from long reads generated by PacBio single-molecule real-time sequencing. Long reads were assembled with Falcon (*Chin et al., 2016*) (version 0.5.4, max_diff = 150, max_cov = 150, min_cov = 2). The resulting primary contig set was iteratively polished with Quiver again using long reads (*Chin et al., 2013*) (version 2.0.0) and with Pilon (*Walker et al., 2014*) (version 1.16) using short reads from a single Illumina TruSeq DNA PCR-free library. The draft genome of *C. orientalis* comprises 135 Mb distributed over 423 gap-free contigs and covers 60% of the *C. rubella* reference with non-ambiguous 1-to-1 whole genome alignments. Its completeness is comparable to that of the *C. rubella* reference.

## Acknowledgements

We thank Christa Lanz for expert assistance with Illumina sequencing. We thank Danelle Seymour, Rebecca Schwab, Beth Rowan, Derek Lundberg, Wangsheng Zhu, Efthymia Symeonidi, Gautam Shirsekar, Rui Wu, Patricia Lang, Talia Karasov, Hernán Burbano, Moisés Exposito Alonso, Maricris Zaidem, Rafal Gutaker, Eunyoung Chae, and Diep Tran for reading of the manuscript and insightful comments. Thank you to Dmitry German for his identification of *C orientalis* from herbarium

samples, making this study possible in the first place. This work was supported by a Human Frontiers Science Program Long-Term Fellowship to DK (LT000783/2010 L) and by DFG-SPP1529 ADAPTOMICS (WE 2897/4–2), ERC Advanced Grant IMMUNEMESIS and the Max Planck Society (DW).

## Additional information

### Competing interests
Detlef Weigel: Deputy editor, *eLife*. The other authors declare that no competing interests exist.

### Funding

| Funder | Grant reference number | Author |
| --- | --- | --- |
| European Research Council | IMMUNEMESIS | Detlef Weigel |
| Human Frontier Science Program | LT000783/2010-L | Daniel Koenig |
| Deutsche Forschungsgemeinschaft | WE 2897/4-2 | Detlef Weigel |
| Max-Planck-Gesellschaft | Open-access funding | Detlef Weigel |

The funders had no role in study design, data collection and interpretation, or the decision to submit the work for publication.

### Author contributions
Daniel Koenig, Conceptualization, Resources, Data curation, Software, Formal analysis, Supervision, Funding acquisition, Validation, Investigation, Visualization, Methodology, Writing—original draft, Project administration, Writing—review and editing; Jörg Hagmann, Data curation, Writing—review and editing; Rachel Li, Felix Bemm, Investigation, Writing—review and editing; Tanja Slotte, Stephen I Wright, Resources, Supervision, Writing—review and editing; Barbara Neuffer, Resources, Writing—review and editing; Detlef Weigel, Conceptualization, Supervision, Funding acquisition, Writing—original draft, Project administration, Writing—review and editing

### Author ORCIDs
Daniel Koenig http://orcid.org/0000-0002-1037-5346
Rachel Li https://orcid.org/0000-0002-8112-4237
Stephen I Wright http://orcid.org/0000-0001-9973-9697
Detlef Weigel http://orcid.org/0000-0002-2114-7963

### Decision letter and Author response
Decision letter https://doi.org/10.7554/eLife.43606.047
Author response https://doi.org/10.7554/eLife.43606.048

## Additional files

### Supplementary files
• Supplementary file 1. GO enrichment analysis of tsSNPs.
DOI: https://doi.org/10.7554/eLife.43606.035

• Supplementary file 2. GO enrichment for $tr_{3-way}$SNPs.
DOI: https://doi.org/10.7554/eLife.43606.036

• Supplementary file 3. List of well covered genes for targeted analysis of potential balancing selection.
DOI: https://doi.org/10.7554/eLife.43606.037

• Supplementary file 4. Pericentromeric or repeat dense genomic regions filtered in genome scans.
DOI: https://doi.org/10.7554/eLife.43606.038

• Supplementary file 5. Annotation hierarchies for SNPs with multiple annotations.
DOI: https://doi.org/10.7554/eLife.43606.039

• Supplementary file 6. List of *A. thaliana* NLR genes used for ortholog identification.
DOI: https://doi.org/10.7554/eLife.43606.040

• Transparent reporting form
DOI: https://doi.org/10.7554/eLife.43606.041

### Data availability

All raw sequencing data are depsoited under the accession codes PRJEB6689.

The following dataset was generated:

| Author(s) | Year | Dataset title | Dataset URL | Database and Identifier |
|---|---|---|---|---|
| Koenig D, Hag-mann J, Li R, Bemm F, Slotte T, Neuffer B, Wright SI, Detlef Weigel | 2018 | Whole genome resequencing of Capsella species | https://www.ebi.ac.uk/ena/data/view/PRJEB6689 | European Nucleotide Archive, PRJEB6689 |

The following previously published dataset was used:

| Author(s) | Year | Dataset title | Dataset URL | Database and Identifier |
|---|---|---|---|---|
| Williamson R, Josephs EB, Platts AE | 2014 | Capsella grandiflora WGS | https://www.ebi.ac.uk/ena/data/view/PRJEB6689 | European Nucleotide Archive, PRJEB6689 |

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
