## [Decision Letter]

Thank you for submitting your article "Long-term balancing selection drives evolution of immunity in *Capsella*" for consideration by *eLife*. Your article has been reviewed by Ian Baldwin as the Senior Editor, a Reviewing Editor and three reviewers. The reviewers have opted to remain anonymous.

The reviewers have discussed the reviews with one another and the Reviewing Editor has drafted this decision to help you prepare a revised submission. As you will see from the detailed comments provided below, the reviewers found the findings quite interesting, but raised a number of important concerns that will require both additional analyses and extensive rewriting of the manuscript.

Since additional analyses are needed, and the authors may want to collect more data to respond to the concern about possible artifacts in *C. orientalis*, it may not be feasible for the revisions to be completed within two months. We leave it to the discretion of the authors whether they think this timeline is feasible, or would prefer to work on revisions at more length and resubmit the manuscript to be considered anew.

In terms of the analysis, the three main points are that:

1) The data for *C. orientalis* seems a bit troubling, and should probably be further inspected for possible artifacts.

2) The hypothesis that balancing selection is currently acting in C. rubella needs to be better supported. In particular, the authors need to exclude the possibility that the observed signatures persisted since the ancestral species. It will also be important for the authors to more clearly explain whether they believe that the high variability is driven by introgression (adaptive or not), ongoing balancing selection (and of what kind…) or both.

3) The analysis would be more interpretable if more of the highlighted regions were investigated individually, rather than always treating all tsSNPs jointly.

In terms of the writing, all reviewers agreed that the paper is unrelated to the origin of sex or mating system evolution, and that these topics are not always discussed accurately. The discussions of these points should be removed. All reviewers also felt that a number of claims were too strong, and relevant literature is missing.

We hope you will find these comments useful in revising your manuscript, should you choose to do so.

*Reviewer #1:*

The idea of looking for balanced polymorphisms in selfers, in which the footprints are frozen by the lower effective recombination rate, is really nice, and I was convinced by the findings that increased diversity and other signs of balancing selection were enriched for host-pathogen genes.

Less clear to me is when/where the balancing selection is occurring. There are at least three possibilities:

1) The hypothesis the authors privilege: long-term balancing selection is ongoing in the selfers as well as the outcrossing species.

2) Balancing selection occurred only in the ancestor of the two selfers, increasing the probability that more than one lineage was retained in the selfer. In a parenthetical comment, they dismiss this possibility for orientalis based on a calculation from Wiuf et al., 2004. They need to spell out their assumptions about Ne etc… in doing so. More importantly, they also need to calculate a similar quantity for rubella. Otherwise, a number of statements in the manuscript seemed overstated (e.g., the last sentence of the Abstract).

3) Balancing selection is operating in the outcrosser and variation was introduced into the selfer by introgression. They entertain this possibility for rubella in the Discussion section. They label it as "adaptive introgression" but it seems to me that it could just as well be neutral, if introgression is more likely to introduce a different haplotype due to balancing selection in the outcrosser.

In this regard, I note that it is not correct that (1) and (3) cannot be distinguished, as they predict different genealogies. Given that the selfers are mostly homozygous at these positions, can they build trees for e.g., 100 regions, and see whether they support (1) or (3)? If not, they should at least explore diversity and divergence levels around the tsSNPs.

More generally, I was confused by what specific models the authors had in mind at a number of points in the manuscript. Because they focus on selfers, their study design seems aimed at picking up balancing selection due to fluctuating selection pressures not heterozygote advantage. Moreover, they repeatedly state that they are looking for a form of selection that favors rare types. Why then expect a positive Tajima's D (subsection “A high density of tsSNPs around immunity-related loci”)? I think a discussion of the mode of balancing selection they expect to detect is missing.

The frequency spectrum for orientalis in Figure 5A seems quite odd. And the observation about the low interchromosomal LD among ts3_SNPs is likewise disconcerting. The explanation the authors provide seems highly unlikely, unless there is rampant epistasis among ts SNPs. If there were, shouldn't the authors also expect to see large footprints of balancing selection in the outcrossers (because haplotypes, not just SNPs, would be retained)?

The Red Queen hypothesis is used to refer to a dizzying array of hypotheses, from the evolution of sex (Introduction) to selection for new combinations of alleles (subsection “Balancing selection over ancient time scales”) to selection for rare alleles (Introduction) to balancing selection (Discussion section), in ways that are often inaccurate. In particular, the Red Queen hypothesis is neither necessary nor sufficient for the maintenance of genetic variation. As Aurelien Tellier and others have shown, it is actually very unlikely that genetic variation would be maintained over long time periods due to simple models of host-pathogen coevolution, especially in small populations (see Tellier et al., 2014 Evolution, for example). This literature should be discussed, especially since the authors are finding so many putative examples.

*Reviewer #2:*

I was very disappointed by this manuscript, as the results are presented so as to appear more exciting than they really are, and parts of the study appear to reflect a lack of understanding. In the first category, I would mention text in the Introduction about the evolution of sex, with which the study does not even remotely deal. In the second category of lack of understanding and/or knowledge of the field and of results already established, I highlight the discussion of outcrossing, which does not mention inbreeding depression, which is surely a factor that ought to be mentioned – I think that it is incorrect to predict that in general outcrossing will be most favored by natural selection when the selective landscape changes frequently, and to imply that pathogen pressure is probably important in maintaining outcrossing. Moreover, the conclusion that balancing selection still operates in the selfers to maintain variation seems unjustified, as other possibilities are not excluded, and the same applies to the conclusion stated at the end of the Abstract that "population longevity in the face of pathogen attack depends on the persistence of ancient genetic variation".

The results may support the much more limited, but still interesting, conclusion that genes involved in defences against pathogens are enriched for cases of balancing selection, in line with the many previously published findings that this class of gene functions shows this kind of behaviour. It is far from clear that there is an "advantage of an expanded footprint of balancing selection after a genetic bottleneck", as the Abstract claims. It is correct that low diversity in a selfer might allow rare high diversity loci to be detected, but selfers that evolved recently from an outcrossing population have the disadvantage that one cannot disentangle selection prior to the selfing state being established from selection more recently, to say nothing of the effect of a bottleneck in increasing the variance of LD and other quantities that might be estimated.

Details of these major concerns are too long for your word limit, so I have emailed them to the editorial office in a separate file, to be passed on to the authors.

My recommendation would be to revise the manuscript extensively to show clearly the evidence for those conclusions that are well supported, and relate them to what is already known.

*Reviewer #3:*

In this manuscript, the authors aim to detect evidence and identify targets of balancing selection by genome-wide search of polymorphisms shared by three Caspella species, two of which experienced genetic bottleneck and transition to self-fertilization independently. The authors resequenced 50 C. rubella, 13 C. grandiflora and 16 C. orientalis accessions and identified millions of single-nucleotide polymorphisms (SNPs). With this dataset, they first characterized the demographic history of these species and, unlike a previous study, detected evidence of gene flow between (eastern) C. rubella and C. grandiflora. Then, they detected evidence of balancing selection enriched in immunity-related genes by focusing on shared polymorphisms between species. 21 genomic regions are highlighted as strong candidates of targets under balancing selection, many of which contain immunity genes with known functions in *A. thaliana*.

This study does a reasonably good job in excluding factors other than ancestral variation that could lead to shared polymorphisms, so the conclusions are convincing to me. I also enjoy reading this manuscript, as it generates several new insights: (1) it substantially increases the number of trans-species polymorphisms that are potentially maintained by balancing selection in plants, providing candidates for follow-up functional studies; (2) it strongly suggests that frequency-dependent selection is a common mechanism underlying ancient balancing selection (as compared to heterozygote advantage); (3) the case of MLO2 underscores the technical difficulty of detecting divergent haplogroups shared between species by short-read sequencing methods.

That said, I have several questions and comments that should be considered before publication:

1) The shared IBD segments and genome-wide D-statistic suggested past and ongoing gene flow between sympatric C. rubella and C. grandiflora. Are there shared IBD segments and significant D-statistic values for the regions with strong evidence of balancing selection? The presence of introgression does not change the conclusion that most tsSNPs are retained ancestral variation, as long as western C. rubella and C. grandiflora still share the polymorphisms, but I am wondering if some of the sharing by eastern C. rubella and C. grandiflora could be attributed to recent gene flow.

2) To exclude the possibility of spurious tsSNPs due to cryptic paralogs, the authors compared the coverage and read concordance. Although the overall distribution of the two metrics looks similar for tsSNPs and ssSNPs, it is not clear whether there could be a few regions with excess coverage or reduced concordance. It will be helpful to plot these two metrics along the chromosome coordinate for the regions with strong evidence for balancing selection.

Another way to rule out the concern of the cryptic paralogs is to compare the proportion of heterozygotes for tsSNPs and ssSNPs (controlling for allele frequency). If some of the tsSNPs are due to cryptic paralogs, the proportion of heterozygotes is expected to be higher than that for ssSNPs.

3) The presence of tsSNPs shared by all three species of Capsella provides strong evidence for balancing selection, but I am slightly concerned by the quality of the *C. orientalis* data because of a couple of puzzling patterns: (A) why is the allele frequency spectrum of ssSNPs in *C. orientalis* multi-modal? (B) Why is the average divergence between the two main Capsella lineages higher near sites adjacent to the three-way tsSNPs?

4) The interchromosomal LD for three-way tsSNPs is significantly lower than that of random polymorphisms in *C. rubella* and *C. grandiflora*. The authors speculated that this might be due to selection promoting new combinations of tsSNPs. However, I found this explanation implausible unless strong and prevalent epistasis is involved. Could the authors run simulations to show that the "reshuffling" hypothesis is able to explain the reduced LD and estimate how much epistasis is required to generate the observed pattern? In addition, why is the interchromosomal LD for three-way tsSNPs significantly higher for that of random polymorphisms in *C. orientalis*? Could this be due to technical issues?

[Editors’ note: what now follows is the decision letter after the authors submitted for further consideration.]

Thank you for resubmitting your work entitled "Long-term balancing selection drives evolution of immunity genes in *Capsella*" for further consideration at *eLife*. Your revised article has been favorably evaluated by Ian Baldwin (Senior Editor), a Reviewing Editor, and two reviewers.

The manuscript has been improved but there are some remaining issues with the presentation and discussion that need to be addressed before acceptance, as outlined below. Notably reviewer 2 made a number of useful suggestions of clarifications that would be helpful in understanding the analyses and results.

*Reviewer #1:*

This manuscript is significantly improved. However, the writing is still rough in several places, and there are some other things that should be improved.

The Abstract suggests that the study aim is "to understand the extent to which natural selection can drive the retention of genetic diversity", meaning sequence diversity. In fact, however, the conclusions relate mainly to whether genes whose sequences suggest that they are involved in resistance to pathogens show footprints of balancing selection, as has been suggested many times, and for which there is some empirical evidence, including from analyses of diversity. This study is therefore an advance mainly in that such genes were not chosen for study and shown to have interesting diversity patterns, but instead they emerged from a genomic analysis for balancing selection, and specifically through analyzing shared polymorphisms.

The revised manuscript concludes that genes showing signals of potential balancing selection show an enrichment for genes involved in interactions with pathogens. I have two comments about this. First, are some of these genes located in clusters in physically small genome regions (as is commonly the case for plant NRR genes)? If so, could this over-estimate the fraction under balancing selection, because one gene is actually under balancing selection, and the others have correlated diversity patterns, due to close linkage? This would seem to be a particular concern for any clusters that are located in regions of low recombination, or low effective recombination rates (which could encompass considerable parts of the genomes of the selfing species studied).

It is far from clear that "self-fertilizing species provide increased sensitivity to detect balancing selection" (as claimed in the Introduction) and indeed the evidence is not compelling, though it is overall consistent with its having been detected in the selfing Capsella species studied. Moreover, only certain kinds of balancing selection could be detected in a selfer, notably situations involving frequency dependence (not heterozygote advantage). This claim should therefore not be made.

Second, the revised manuscript makes much clearer than before that the long-term balancing selection that may have been detected could potentially occur solely within the outcrossing species from which the selfing species evolved (in one case, apparently very recently). The signal within the selfing species might therefore not reflect ongoing selection in those species, but merely introgression of sequences bearing the footprints of the selection. The revised text marshalls the evidence on this question quite clearly. However, it is not mentioned that introgression by pollen flow into the selfing species is more likely than the reverse, following the long-established general rule for attempts to inter-cross two species when only one of them is self-incompatible: pollen of the self-compatible species often fails to work on pistils of the self-incompatible one, whereas the reciprocal pollination succeeds (as might be expected if selfing species' pistils have lost the ability to reject incompatible pollen). However, some recently evolved selfing species also reject pollen from their close self-incompatible relatives. A comment on the case in these Capsella species would be worth adding. In my opinion, the abstract should mention that tests for adaptive introgression suggest that this is not causing the signal(s) on which the conclusion of balancing selection in the selfers is based.

The first paragraph is poorly written and somewhat misleading, as it fails to mention that loss of alleles in situation with strong balancing selection occurs only in special cases. I think that the authors meaning is as follows. The term "balancing selection" describes several different adaptive forces that maintain genetic variation for longer than expected under genetic drift of neutral variants. It includes situations with heterozygous advantage, negative frequency-dependent selection (rare allele advantage), and environmental heterogeneity affecting fitnesses in space and time. In these situations, selection prevents loss of alleles from populations at the functional genes or sites. As this also results in increased diversity at closely linked neutral or weakly selected sites (Charlesworth, 2006), it should be possible to detect balancing selection from the resulting footprints of increased coalescence times at closely linked sites, and many candidate genes have been identified using diverse methodology (Fijarczyk and Babik, 2015). However, theoretical models predict that even strongly balanced functional alleles can be stochastically lost over long time periods, suggesting that balanced polymorphism could often be short lived [a reference is needed here], particularly when the functional alleles have low equilibrium frequencies, or fluctuate in frequency, with periods of low frequency, as may occur in the case of plant pathogen resistance genes where temporarily rare alleles may have advantages (Tellier et al., 2014). There is no need to repeat this reference in the Introduction. It would be better to make clear all along that this is a special case.

*Reviewer #2:*

The manuscript has been substantially improved after extensive revisions. Removal of the discussion of evolution of sex makes the main story much clearer, and I appreciate the explicit discussion of different possible mechanisms underlying the shared polymorphisms (technical artifacts, recurrent mutations, maintenance of ancestral variation, introgression, etc). The evidence of trans-species polymorphisms in MLO2b is very compelling, and I am convinced that the high-quality three-way tsSNPs show evidence for long-term balancing selection and no evidence of recent gene flow.

However, I have a few questions about the analysis and hope the authors could add some details in the manuscript to improve clarity. I also feel the current abstract is too vague and lacks specifics of some key results. For instance, it is unclear which collection of variants the authors are referring to when saying "ancestral variation preferentially persists at immunity related loci" (ts_2waySNPs, Bal regions, ts_3waySNPs, or ts_3wayhqSNPs?). In addition, the gene name MLO2b should be specified in the second to the last sentence, as this is the only case the authors demonstrate trans-species sharing of haplotypes clearly. Please see below for my specific questions and comments.

1) The authors use the D statistic and IBD sharing to support their conclusion of ongoing gene flow between C. grandiflora and eastern C rubella, but some details of these analyses are unclear.

D-statistic calculation:

1) What exactly is the configuration for the ABBA-BABA test. For example, what species is used as the outgroup?

2) Why is the correlation between D statistic and distance evidence for ongoing gene flow (as claimed in subsection “*Capsella rubella* demography”)? Could this reflect past gene flow?

IBD detection:

1) What does the minimum segment length of 1kb translate into genetic distance? I am not familiar with recombination rates in plants, but such segments seem very short by standards in humans and thus very old. (The length threshold is only specified in legend of Figure 2, but I think it should also be included in the Materials and methods section.)

2) What is the distribution of detected IBD segment lengths?

3) Can the authors provide some estimates for the age of the IBD segments based on their lengths to support their conclusion of "very recent co-ancestry"?

2) Figure 4 is one of the key results showing enrichment in immunity-related genes, but it is unclear what each panel shows exactly, as some details or labels are missing.

1) Panel A: What does each dot mean? I guess each point represents a gene, but this is not clear from the text, figure or legend. What is the y-axis in each sub-panel? How is this number calculated?

2) Panels B: what do the thick line and the shaded area mean, respectively? Does the legend of panel C apply to panel B?

3) The authors use simulation data to argue against the possibility that the diversity patterns near Bal loci result from historical balancing selection in the ancestor population. Did the simulations include effects of balancing selection? If so, how is balancing selection simulated in practice? If balancing selection is not simulated, are the results based on neutral simulations sufficient to rule out the possibility of historical balancing selection?

4) MLO2b region

1) Why is the haplotype structure of MLO2b not discussed on C grandiflora? Since the tsSNPs are shared by C grandiflora, it seems the selection pressure is preserved in this species. Would the higher recombination rate in C grandiflora help narrow down the causal variant(s)?

2) How long are the shared haplotypes between C rubella and C orientalis? Is maintenance of such long haplotypes expected, given plausible split time and recombination rate, if there is no epistasis? It will be helpful to estimate this using the formula in Wiuf et al., 2004.

3) Based on Figure 3B, haplogroup C differs by only 1 SNP from the second haplotype in group B. Why is C classified as the separate haplogroup instead of a subtype of B?

4) In subsection “Insights into balancing selection from de novoassembly of *MLO2”*, the authors concluded that the shared haplotypes did not result from recent gene flow, because of similar within-haplotype divergence levels of A and B haplogroups. However, this evidence is insufficient: the authors need to demonstrate that the within-haplogroup divergence is at least as high as the genome-wide average (which seems to be the case by comparing Figure 6—figure supplement 1 to Figure 5D).

5) The reconstructed trees in Figure 6C are very small, and haplotypes need to be labeled in order to show the pattern of clustering by haplogroup.

---

## [Author Response]

[Editors’ note: the author responses to the first round of peer review follow.]

Since additional analyses are needed, and the authors may want to collect more data to respond to the concern about possible artifacts in C. orientalis, it may not be feasible for the revisions to be completed within two months. We leave it to the discretion of the authors whether they think this timeline is feasible, or would prefer to work on revisions at more length and resubmit the manuscript to be considered anew.In terms of the analysis, the three main points are that:1) The data for C. orientalis seems a bit troubling, and should probably be further inspected for possible artifacts.

The reviewers main concern regarding the *C. orientalis* data was that the allele frequency spectrum was unusual which may indicate elevated error rates. The unusual appearance of the spectrum is the result of having 16 samples, and binning allele frequencies into 10 bins. Because we had differing numbers of individuals from each species, we chose to compare everything using the same binning, which happens to mathematically introduce an uneven spectrum as observed in the original Figure 5. We replotted these spectra using different bin sizes for each species and show that the uneven pattern is largely removed.

To further alleviate the reviewers’ concerns, we generated a draft genome assembly for one of our *C. orientalis* samples using an alternative sequencing technology and recalled variants at tsSNP sites in our *C. orientalis samples*. In the updated manuscript, only SNPs identified in both approaches were considered “high quality”.

2) The hypothesis that balancing selection is currently acting in C. rubella needs to be better supported. In particular, the authors need to exclude the possibility that the observed signatures persisted since the ancestral species.

The reviewers questioned our assertion that balancing selection has retained genetic diversity in selfing lineages and suggested that the historical molecular signature of balancing selection in the outcrossing *C. grandiflora*, might result in the appearance of balancing selection in *C. rubella*. Specifically, if immunity related genes are under balancing selection in the outcrosser, then a random sample of alleles during the *C. rubella* founding event might result in higher genetic variation in subsequent generations.

We agree with the reviewers that these two possibilities are difficult to tease apart in the recent *C. rubella / C. grandiflora* speciation event. We based our original conclusion that balancing selection is operating in *C. rubella* based on the following arguments:

1) The allele frequencies in the proposed regions are far less diverged between *C. grandiflora* and *C. rubella* then expected by random chance under an explicit demographic model which includes gene flow.

2) These regions show many classic indications of balancing selection in *C. rubella* including:

a) Increased allele frequency

b) Increased genetic diversity

c) Low Fst between subpopulations__

d) Elevated Tajima’s D

3) In *C. grandiflora,* genetic diversity is high throughout the genome, and balanced regions show only subtle deviation from the global pattern very near to the candidate targets. Therefore, a random sample of alleles anywhere in the genome would generate peaks of outstandingly high genetic diversity in *C. rubella*, yet we see that retention of variation is specifically elevated at immunity loci.

4) Though we identified tsSNPs throughout the genome, those tsSNPs found in our candidate regions were much more likely to be retained after population splits regardless of allele frequency.

5) In the much more diverged and much less diverse selfing lineage, *C. orientalis*, we observe the same pattern at many of the same loci. Though this is not direct evidence for balancing selection in *C. rubella* specifically, it is strong evidence that these loci are the targets of stable, long-term balancing selection in a selfing lineage. It is possible that, despite having been stable over millions of years of evolution previously, balancing selection ended within the last ~150,000 years in *C. rubella* specifically. We think that the more plausible explanation of the observed patterns is that variation is retained by ongoing selection in all three lineages.

In response to this concern we sought to strengthen our argument with two additional analyses. First, we examined whether elevated founder diversity might impact the likelihood of observing the pattern of Fst we used to identify outliers. We approximated the effect of balancing selection in *C. grandiflora* alone by including only simulations that showed the highest level of diversity *C. grandiflora*, and then explored whether our outlier *C. rubella* Fst values were more likely to occur. Even under this scenario of highfounder diversity, the observed values remain extremely unlikely. Second, we examined whether allele frequencies are correlated between *C. grandiflora* and *C. rubella* and between subpopulations of *C. rubella* for tsSNPs inside and outside of balanced regions.

We find that although all these SNPs segregate in all of the populations assayed, SNPs within balanced regions are show a substantial increase in AFS correlation.

It will also be important for the authors to more clearly explain whether they believe that the high variability is driven by introgression (adaptive or not), ongoing balancing selection (and of what kind…) or both.

We have added to discussion to make our thinking more clear on these issues. We do not think that adaptive introgression and maintenance by balancing selection are mutually exclusive hypotheses, as the later is not dependent on the source of an allele. In the case of a young species like *C. rubella* we think that gene flow has contributed to the modern pattern of variation, and that this variation is maintained for longer than expected by chance in regions near immunity related loci. For the older split, we find no evidence for recent introgression.

We further explored these ideas by looking for IBD regions inside and outside of balanced regions and find a pattern consistent with the above hypothesis.

With respect to the mechanism of balancing selection, we believe that definitive statements about this issue are beyond the scope of the current manuscript, however we have tried to make it clear that we think heterozygote advantage is unlikely.

3) The analysis would be more interpretable if more of the highlighted regions were investigated individually, rather than always treating all tsSNPs jointly.

We have reported statistics individually for these regions.

In terms of the writing, all reviewers agreed that the paper is unrelated to the origin of sex or mating system evolution, and that these topics are not always discussed accurately. The discussions of these points should be removed.

We have removed this emphasis throughout the manuscript.

All reviewers also felt that a number of claims were too strong, and relevant literature is missing.

We have softened claims that were pointed out by the reviewers as too strong and added relevant literature where requested.

We hope you will find these comments useful in revising your manuscript, should you choose to do so.

Reviewer #1:

The idea of looking for balanced polymorphisms in selfers, in which the footprints are frozen by the lower effective recombination rate, is really nice, and I was convinced by the findings that increased diversity and other signs of balancing selection were enriched for host-pathogen genes.

Thank you for the positive feedback.

Less clear to me is when/where the balancing selection is occurring. There are at least three possibilities:1) The hypothesis the authors privilege: long-term balancing selection is ongoing in the selfers as well as the outcrossing species.2) Balancing selection occurred only in the ancestor of the two selfers, increasing the probability that more than one lineage was retained in the selfer. In a parenthetical comment, they dismiss this possibility for orientalis based on a calculation from Wiuf et al., 2004. They need to spell out their assumptions about Ne etc… in doing so.

We have added a comment on the Ne assumption for the *C. orientalis* calculation.

More importantly, they also need to calculate a similar quantity for rubella. Otherwise, a number of statements in the manuscript seemed overstated (e.g., the last sentence of the Abstract).

We are unsure whether the reviewer is suggesting that we calculate this probability for C. rubella-C. grandiflora or C. rubella-C. orientalis. The C. grandiflora-C. orientalis calculation is conservative relative to the later. However, we have modified the Abstract extensively to account for the reviewer’s criticism.

3) Balancing selection is operating in the outcrosser and variation was introduced into the selfer by introgression. They entertain this possibility for rubella in the Discussion section. They label it as "adaptive introgression" but it seems to me that it could just as well be neutral, if introgression is more likely to introduce a different haplotype due to balancing selection in the outcrosser.

We refer the reviewer to our response to the general comments.

In this regard, I note that it is not correct that (1) and (3) cannot be distinguished, as they predict different genealogies. Given that the selfers are mostly homozygous at these positions, can they build trees for e.g., 100 regions, and see whether they support (1) or (3)? If not, they should at least explore diversity and divergence levels around the tsSNPs.

We have removed this statement and have included additional analysis of IBS regions to address the reviewer’s comment, and we present trees as suggested in our analysis of *MLO2* where complete data on a single gene could be produced.

More generally, I was confused by what specific models the authors had in mind at a number of points in the manuscript. Because they focus on selfers, their study design seems aimed at picking up balancing selection due to fluctuating selection pressures not heterozygote advantage. Moreover, they repeatedly state that they are looking for a form of selection that favors rare types. Why then expect a positive Tajima's D (subsection “A high density of tsSNPs around immunity-related loci”)? I think a discussion of the mode of balancing selection they expect to detect is missing.

We have heavily modified our introduction and removed some of the references that the reviewer refers to. With respect to negative frequency dependent selection, though this selection favors rare alleles, it is expected to elevate Tajima’s D as observed for the classic S-locus case. We have also made specific wording to with respect to the different types of balancing selection that might be operating in the Discussion section.

The frequency spectrum for orientalis in Figure 5A seems quite odd.

Please see the response to the general comments.

And the observation about the low interchromosomal LD among ts3_SNPs is likewise disconcerting. The explanation the authors provide seems highly unlikely, unless there is rampant epistasis among ts SNPs. If there were, shouldn't the authors also expect to see large footprints of balancing selection in the outcrossers (because haplotypes, not just SNPs, would be retained)?

We agree and have removed these data/interpretation.

The Red Queen hypothesis is used to refer to a dizzying array of hypotheses, from the evolution of sex (Introduction) to selection for new combinations of alleles (subsection “Balancing selection over ancient time scales”) to selection for rare alleles (Introduction) to balancing selection (Discussion section), in ways that are often inaccurate. In particular, the Red Queen hypothesis is neither necessary nor sufficient for the maintenance of genetic variation. As Aurelien Tellier and others have shown, it is actually very unlikely that genetic variation would be maintained over long time periods due to simple models of host-pathogen coevolution, especially in small populations (see Tellier et al., 2014 Evolution, for example). This literature should be discussed, especially since the authors are finding so many putative examples.

We have removed the Red Queen references in our work. We have also added reference to Tellier.

Reviewer #2:

I was very disappointed by this manuscript, as the results are presented so as to appear more exciting than they really are, and parts of the study appear to reflect a lack of understanding.

We did not mean to present our results as more exciting than they are, only as exciting as we found them. We hope that the reviewer is more satisfied with our revised presentation.

In the first category, I would mention text in the Introduction about the evolution of sex, with which the study does not even remotely deal. In the second category of lack of understanding and/or knowledge of the field and of results already established, I highlight the discussion of outcrossing, which does not mention inbreeding depression, which is surely a factor that ought to be mentioned – I think that it is incorrect to predict that in general outcrossing will be most favored by natural selection when the selective landscape changes frequently, and to imply that pathogen pressure is probably important in maintaining outcrossing.

We have removed all of this material from the manuscript.

Moreover, the conclusion that balancing selection still operates in the selfers to maintain variation seems unjustified, as other possibilities are not excluded, and

We have added several additional analyses supporting our conclusions.

the same applies to the conclusion stated at the end of the Abstract that "population longevity in the face of pathogen attack depends on the persistence of ancient genetic variation".

We have revised the Abstract and modified this statement.

The results may support the much more limited, but still interesting, conclusion that genes involved in defences against pathogens are enriched for cases of balancing selection, in line with the many previously published findings that this class of gene functions shows this kind of behaviour. It is far from clear that there is an "advantage of an expanded footprint of balancing selection after a genetic bottleneck", as the Abstract claims. It is correct that low diversity in a selfer might allow rare high diversity loci to be detected, but selfers that evolved recently from an outcrossing population have the disadvantage that one cannot disentangle selection prior to the selfing state being established from selection more recently, to say nothing of the effect of a bottleneck in increasing the variance of LD and other quantities that might be estimated.

Please see our response to the general editorial comments.

Details of these major concerns are too long for your word limit, so I have emailed them to the editorial office in a separate file, to be passed on to the authors.My recommendation would be to revise the manuscript extensively to show clearly the evidence for those conclusions that are well supported, and relate them to what is already known.

Reviewer #3:

[…] This study does a reasonably good job in excluding factors other than ancestral variation that could lead to shared polymorphisms, so the conclusions are convincing to me. I also enjoy reading this manuscript, as it generates several new insights: (1) it substantially increases the number of trans-species polymorphisms that are potentially maintained by balancing selection in plants, providing candidates for follow-up functional studies; (2) it strongly suggests that frequency-dependent selection is a common mechanism underlying ancient balancing selection (as compared to heterozygote advantage); (3) the case of MLO2 underscores the technical difficulty of detecting divergent haplogroups shared between species by short-read sequencing methods.

We appreciate the reviewers’ positive comments regarding the manuscript.

That said, I have several questions and comments that should be considered before publication:1) The shared IBD segments and genome-wide D-statistic suggested past and ongoing gene flow between sympatric C. rubella and C. grandiflora. Are there shared IBD segments and significant D-statistic values for the regions with strong evidence of balancing selection? The presence of introgression does not change the conclusion that most tsSNPs are retained ancestral variation, as long as western C. rubella and C. grandiflora still share the polymorphisms, but I am wondering if some of the sharing by eastern C. rubella and C. grandiflora could be attributed to recent gene flow.

We have added analyses of IBD segments along the lines of the reviewers’ suggestions.

2) To exclude the possibility of spurious tsSNPs due to cryptic paralogs, the authors compared the coverage and read concordance. Although the overall distribution of the two metrics looks similar for tsSNPs and ssSNPs, it is not clear whether there could be a few regions with excess coverage or reduced concordance. It will be helpful to plot these two metrics along the chromosome coordinate for the regions with strong evidence for balancing selection. Another way to rule out the concern of the cryptic paralogs is to compare the proportion of heterozygotes for tsSNPs and ssSNPs (controlling for allele frequency). If some of the tsSNPs are due to cryptic paralogs, the proportion of heterozygotes is expected to be higher than that for ssSNPs.

We decided that this suggestion was the simplest way of addressing this concern and have provided it in the manuscript. We modified this approach a bit, we checked to see if the balanced regions that show signal of increased heterozygosity, rather than tsSNPs. We find no elevations of observed relative to expected heterozygosity in these regions.

3) The presence of tsSNPs shared by all three species of Capsella provides strong evidence for balancing selection, but I am slightly concerned by the quality of the C. orientalis data because of a couple of puzzling patterns: (A) why is the allele frequency spectrum of ssSNPs in C. orientalis multi-modal?

Please see the discussion of this issue in the general comments.

B) Why is the average divergence between the two main Capsella lineages higher near sites adjacent to the three-way tsSNPs?

We assume that the reviewer means relative to the genome-wide average. In our manuscript, we hypothesize that these alleles have been segregating in *Capsella* since before the species split. Thus, on average, we expect divergence to be slightly elevated in tight linkage with the balanced SNP (since the exact same allele was not selected in each lineage, but just related alleles), and this is indeed what we see.

4) The interchromosomal LD for three-way tsSNPs is significantly lower than that of random polymorphisms in C. rubella and C. grandiflora. The authors speculated that this might be due to selection promoting new combinations of tsSNPs. However, I found this explanation implausible unless strong and prevalent epistasis is involved. Could the authors run simulations to show that the "reshuffling" hypothesis is able to explain the reduced LD and estimate how much epistasis is required to generate the observed pattern? In addition, why is the interchromosomal LD for three-way tsSNPs significantly higher for that of random polymorphisms in C. orientalis? Could this be due to technical issues?

We have removed this analysis from the manuscript.

[Editors' note: the author responses to the re-review follow.]

The manuscript has been improved but there are some remaining issues with the presentation and discussion that need to be addressed before acceptance, as outlined below. Notably reviewer 2 made a number of useful suggestions of clarifications that would be helpful in understanding the analyses and results.

Reviewer #1:

[…] The revised manuscript concludes that genes showing signals of potential balancing selection show an enrichment for genes involved in interactions with pathogens. I have two comments about this. First, are some of these genes located in clusters in physically small genome regions (as is commonly the case for plant NRR genes)? If so, could this over-estimate the fraction under balancing selection, because one gene is actually under balancing selection, and the others have correlated diversity patterns, due to close linkage? This would seem to be a particular concern for any clusters that are located in regions of low recombination, or low effective recombination rates (which could encompass considerable parts of the genomes of the selfing species studied).

We agree with the reviewer’s comments, as we noted in the discussion “It is possible, or perhaps even likely, that the signal of balancing selection is amplified by the fact that immunity-related loci occur in clusters (Meyers et al., 2003) and that our strongest signal is the result of simultaneous selection on several genes in these regions in a situation analogous to the MHC in animals (Hedrick, 1998). Thus, biotic factors might not be quite as important as our analyses make them appear.” We hope that the reviewer finds this recognition of the biases of gene clusters acceptable.

It is far from clear that "self-fertilizing species provide increased sensitivity to detect balancing selection" (as claimed in the Introduction) and indeed the evidence is not compelling, though it is overall consistent with its having been detected in the selfing Capsella species studied. Moreover, only certain kinds of balancing selection could be detected in a selfer, notably situations involving frequency dependence (not heterozygote advantage). This claim should therefore not be made.

The beginning of the sentence actually reads “We hypothesized that self-fertilizing…”. We feel that we clearly state a hypothesis, not a conclusion. We hope we can leave it to the reader to decide whether this particular idea has merit, and indeed, we point out many of the caveats that the reviewer mentions throughout the manuscript. We think that our study does provide some evidence for this hypothesis, and we hope to learn if it is widely true from further studies.

Second, the revised manuscript makes much clearer than before that the long-term balancing selection that may have been detected could potentially occur solely within the outcrossing species from which the selfing species evolved (in one case, apparently very recently). The signal within the selfing species might therefore not reflect ongoing selection in those species, but merely introgression of sequences bearing the footprints of the selection. The revised text marshalls the evidence on this question quite clearly. However, it is not mentioned that introgression by pollen flow into the selfing species is more likely than the reverse, following the long-established general rule for attempts to inter-cross two species when only one of them is self-incompatible: pollen of the self-compatible species often fails to work on pistils of the self-incompatible one, whereas the reciprocal pollination succeeds (as might be expected if selfing species' pistils have lost the ability to reject incompatible pollen). However, some recently evolved selfing species also reject pollen from their close self-incompatible relatives. A comment on the case in these Capsella species would be worth adding.

We have added a comment on the direction of fertility and a reference that explores the hybrid compatibility of these species in detail.

In my opinion, the abstract should mention that tests for adaptive introgression suggest that this is not causing the signal(s) on which the conclusion of balancing selection in the selfers is based.

We preferred to leave this out of the abstract because it is entirely possible that adaptive introgression plays an important role during the early stages of divergence. It is true that recent gene flow between *C. orientalis* and the other species has not occurred (as stated in the text), but we thought that including a broader statement might confuse the reader.

The first paragraph is poorly written and somewhat misleading, as it fails to mention that loss of alleles in situation with strong balancing selection occurs only in special cases. I think that the authors meaning is as follows. The term "balancing selection" describes several different adaptive forces that maintain genetic variation for longer than expected under genetic drift of neutral variants. It includes situations with heterozygous advantage, negative frequency-dependent selection (rare allele advantage), and environmental heterogeneity affecting fitnesses in space and time. In these situations, selection prevents loss of alleles from populations at the functional genes or sites. As this also results in increased diversity at closely linked neutral or weakly selected sites (Charlesworth, 2006), it should be possible to detect balancing selection from the resulting footprints of increased coalescence times at closely linked sites, and many candidate genes have been identified using diverse methodology (Fijarczyk and Babik, 2015). However, theoretical models predict that even strongly balanced functional alleles can be stochastically lost over long time periods, suggesting that balanced polymorphism could often be short lived [a reference is needed here], particularly when the functional alleles have low equilibrium frequencies, or fluctuate in frequency, with periods of low frequency, as may occur in the case of plant pathogen resistance genes where temporarily rare alleles may have advantages (Tellier et al., 2014). There is no need to repeat this reference in the Introduction. It would be better to make clear all along that this is a special case.

We agree that the use of the Tellier reference was unclear, as it does support the possibility of identifying balanced sites, the Fijarczyk review covers the broad consensus that balancing selection over long time scales is rare. The second reference to Tellier has been removed.

Reviewer #2:

The manuscript has been substantially improved after extensive revisions. Removal of the discussion of evolution of sex makes the main story much clearer, and I appreciate the explicit discussion of different possible mechanisms underlying the shared polymorphisms (technical artifacts, recurrent mutations, maintenance of ancestral variation, introgression, etc). The evidence of trans-species polymorphisms in MLO2b is very compelling, and I am convinced that the high-quality three-way tsSNPs show evidence for long-term balancing selection and no evidence of recent gene flow.However, I have a few questions about the analysis and hope the authors could add some details in the manuscript to improve clarity.

We thank the reviewer for their comments, and we hope that the changes documented below will satisfy their concerns.

I also feel the current abstract is too vague and lacks specifics of some key results. For instance, it is unclear which collection of variants the authors are referring to when saying "ancestral variation preferentially persists at immunity related loci" (ts_2waySNPs, Bal regions, ts_3waySNPs, or ts_3wayhqSNPs?).

All of the SNP classes suggested by the reviewer were enriched in immune loci. We feel that each subset is better defined in the text itself.

In addition, the gene name MLO2b should be specified in the second to the last sentence, as this is the only case the authors demonstrate trans-species sharing of haplotypes clearly.

We have added a sentence specifically in reference to *MLO2* to the abstract. We note that ts3wayhqSNPS were found to be in LD with at least one other ts3wasyhqSNP in all three species.

Please see below for my specific questions and comments.1) The authors use the D statistic and IBD sharing to support their conclusion of ongoing gene flow between C. grandiflora and eastern C rubella, but some details of these analyses are unclear.D-statistic calculation:1) What exactly is the configuration for the ABBA-BABA test. For example, what species is used as the outgroup?

This information has been added to the Materials and methods section.

2) Why is the correlation between D statistic and distance evidence for ongoing gene flow (as claimed in subsection “Capsella rubella demography”)? Could this reflect past gene flow?

The word “ongoing” has been removed.

IBD detection:1) What does the minimum segment length of 1kb translate into genetic distance? I am not familiar with recombination rates in plants, but such segments seem very short by standards in humans and thus very old. (The length threshold is only specified in legend of Figure 2, but I think it should also be included in the Materials and methods section.)

The effective population size of *C. grandiflora* is several orders of magnitude larger than humans, meaning that effective recombination rate is much higher. IBD segments of the length observed in humans are very unlikely in this context (and shorter segments are supported by more SNPs than would be expected in human genetic data). The length threshold has been added to the Materials and methods section.

2) What is the distribution of detected IBD segment lengths?

This data is now shown in Figure 4—figure supplement 4.

3) Can the authors provide some estimates for the age of the IBD segments based on their lengths to support their conclusion of "very recent co-ancestry"?

We changed “very” to “more”. We think that estimating the age of IBD segments might be possible, however this would not be trivial given the differing recombination rates and population structure of our sample. Since our goal is not to estimate the rate of gene flow accurately, but to show that balancing selection is acting in the selfer regardless of the origin of the allele, we feel that this analysis is beyond the scope of the current work.

2) Figure 4 is one of the key results showing enrichment in immunity-related genes, but it is unclear what each panel shows exactly, as some details or labels are missing.1) Panel A: What does each dot mean? I guess each point represents a gene, but this is not clear from the text, figure or legend. What is the y-axis in each sub-panel? How is this number calculated?

To clarify the figure, we have added the following text to the figure legend “Each point represents values calculated for an individual gene. For example, in the upper subplot each point is the number of tsSNPs identified in a gene divided by the total number of SNPs identified for a gene”.

2) Panel B: what do the thick line and the shaded area mean, respectively? Does the legend of panel C apply to panel B?

The text “For (B-C) the…” has been added to clarify the legend.

3) The authors use simulation data to argue against the possibility that the diversity patterns near Bal loci result from historical balancing selection in the ancestor population. Did the simulations include effects of balancing selection? If so, how is balancing selection simulated in practice? If balancing selection is not simulated, are the results based on neutral simulations sufficient to rule out the possibility of historical balancing selection?

The principal effect of balancing selection, increased diversity, was approximated in simulations by recalculating the probability of the observed values in the selfer using only simulations that showed outstandingly high diversity in *C. grandiflora*. The observed values could not be explained under these conditions.

We acknowledge that this is only an approximation of balancing selection. We note that the reviewer’s previous concerns were that increased founding diversity due to balancing selection could potentially increase the probability of sampling multiple haplotypes during speciation of *C. rubella*. We feel that this specific concern is well addressed by our method.

Still, we recognize that it is difficult to prove beyond all doubts that balancing selection is ongoing in *C. rubella*. Even if we were to explicitly model balancing selection in these two species, it would be difficult if not impossible to consider all possible relevant scenarios. We try to illustrate this challenge throughout the manuscript. For example: “Although evidence for balancing selection at immunity-related loci in *C. rubella* is much stronger than in *C. grandiflora*, it is difficult to completely exclude the effect of founder diversity at these loci on the observed patterns.” This was exactly the reason we analyzed *C. orientalis* and found considerable additional evidence that the same regions were targets of balancing selection in a much older selfer.

4) MLO2b region1) Why is the haplotype structure of MLO2b not discussed on C grandiflora? Since the tsSNPs are shared by C grandiflora, it seems the selection pressure is preserved in this species. Would the higher recombination rate in C grandiflora help narrow down the causal variant(s)?

Our data is unphased Illumina short read data. We can assemble high confidence haplotypes in the sellers because they have very low observed heterozygosity. Accurate haplotype assembly in *C. grandiflora* might be possible with a larger sample size, and it is possible this data might narrow down causal variants. However, it is important to point out that the *C. rubella* and *C. grandiflora* haplotypes have had nearly identical amount of time to recombine since the split with *C. orientalis* (the shift to selfing in *C. rubella* is much more recent in comparison to the larger divergence).

2) How long are the shared haplotypes between C rubella and C orientalis? Is maintenance of such long haplotypes expected, given plausible split time and recombination rate, if there is no epistasis? It will be helpful to estimate this using the formula in Wiuf et al., 2004.

This is a very interesting question. Intuitively, it does not seem probable that now recombination has occurred. Epistasis is a fantastic hypothesis, and one that would certainly be worth exploring in the future. Another possibility is that local variation in recombination rate, potentially further reduced by divergence, is involved. Using the Wiuf equation with an extremely rough estimate of recombination rate does not seem to distinguish these two possible causal factors.

3) Based on Figure 3B, haplogroup C differs by only 1 SNP from the second haplotype in group B. Why is C classified as the separate haplogroup instead of a subtype of B?

We initially separated haplogroup C because the internal four SNPs were in a different orientation then observed in haplogroup A and B, and this orientation was not found to be transpecific. We admit that this was somewhat arbitrary. As it turns out, examination of the assembled sequences indicates that this was the correct decision, this haplogroup was found only in *C. rubella* and is quite diverged from A or B. However, only A and B were clearly transpecific. Because we did not find C in both species, we did not include it in the coverage analysis.

To improve the clarity of the relationships between haplogroups, we have included C in the phylogeny in Figure 6—figure supplement 2.

4) In subsection “Insights into balancing selection from de novo assembly of MLO2”, the authors concluded that the shared haplotypes did not result from recent gene flow, because of similar within-haplotype divergence levels of A and B haplogroups. However, this evidence is insufficient: the authors need to demonstrate that the within-haplogroup divergence is at least as high as the genome-wide average (which seems to be the case by comparing Figure 6—figure supplement 1 to Figure 5D).

We have modified the text to reflect this fact.

5) The reconstructed trees in Figure 6C are very small, and haplotypes need to be labeled in order to show the pattern of clustering by haplogroup.

We agree that the small trees in Figure 6 make it difficult to clearly show all of the haplotype relationships. We have provided a full maximum likelihood tree with all the assembled haplotypes as a separate supplemental figure.